# Stem cell topography splits growth and homeostatic functions in the fish gill

Julian Stolper[1,2], Elizabeth Mayela Ambrosio[1†], Diana-Patricia Danciu[3], Lorena Buono[4], David A Elliott[2], Kiyoshi Naruse[5], Juan R Martínez-Morales[4], Anna Marciniak-Czochra[3,6], Lazaro Centanin[1]*

[1]Centre for Organismal Studies, Heidelberg University, Heidelberg, Germany; [2]Murdoch Children's Research Institute, Royal Children's Hospital, Parkville, Australia; [3]Institute of Applied Mathematics, Heidelberg University, Heidelberg, Germany; [4]Centro Andaluz de Biología del Desarrollo, Universidad Pablo de Olavide, Seville, Spain; [5]Laboratory of Bioresources, National Institute for Basic Biology, National Institutes of Natural Sciences, Okazaki, Japan; [6]Bioquant Center, Heidelberg University, Heidelberg, Germany

*For correspondence:
lazaro.centanin@cos.uni-heidelberg.de

Present address: †The Novo Nordisk Foundation Center for Stem Cell Research, University of Copenhagen, Copenhagen, Denmark

Competing interests: The authors declare that no competing interests exist.

**Abstract** While lower vertebrates contain adult stem cells (aSCs) that maintain homeostasis and drive un-exhaustive organismal growth, mammalian aSCs display mainly the homeostatic function. Here, we use lineage analysis in the medaka fish gill to address aSCs and report separate stem cell populations for homeostasis and growth. These aSCs are fate-restricted during the entire post-embryonic life and even during re-generation paradigms. We use chimeric animals to demonstrate that *p53* mediates growth coordination among fate-restricted aSCs, suggesting a hierarchical organisation among lineages in composite organs like the fish gill. Homeostatic and growth aSCs are clonal but differ in their topology; modifications in tissue architecture can convert the homeostatic zone into a growth zone, indicating a leading role for the physical niche defining stem cell output. We hypothesise that physical niches are main players to restrict aSCs to a homeostatic function in animals with fixed adult size.
DOI: https://doi.org/10.7554/eLife.43747.001

## Introduction

Higher vertebrates acquire a definitive body size around the time of their sexual maturation. Although many adult stem cells (aSCs) remain active and keep producing new cells afterwards, they mainly replace cells that are lost on a daily basis. On the other hand, lower vertebrates like fish keep increasing their size even during adulthood due to the capacity of aSCs to drive growth in parallel to maintaining organ homeostasis. The basis for the different outputs between aSCs in lower and higher vertebrates is still not fully understood. It has been reported, however, that in pathological conditions mammalian aSCs exhibit the ability to drive growth, as best represented by cancer stem cells (CSCs) (*Batlle and Clevers, 2017*; *Nassar and Blanpain, 2016*; *Clevers, 2011*; *Suvà et al., 2014*; *Quintana et al., 2008*; *Barker et al., 2009*; *Schepers et al., 2012*; *Boumahdi et al., 2014*).

Since stem cells in fish maintain homeostasis and drive post-embryonic growth in a highly controlled manner, the system permits identifying similarities and differences in case both functions are performed by dedicated populations or identifying specific conditions within a common stem cell pool driving homeostasis and growth. There are several genetic tools and techniques to explore aSCs in fish, and an abundant literature covering different aspects of their biology in various organs and also during regeneration paradigms (*Gupta and Poss, 2012*; *Knopf et al., 2011*; *Tu and Johnson, 2011*; *Kizil et al., 2012*; *Kyritsis et al., 2012*; *Pan et al., 2013*; *Centanin et al., 2014*; *Jungke et al., 2015*; *Henninger et al., 2017*; *Singh et al., 2017*; *McKenna et al., 2016*;

*Aghaallaei et al., 2016*). Despite all these major advances, we still do not understand whether the same pool of stem cells is responsible for driving both growth and homeostatic replacement, or if alternatively, each task is performed by dedicated aSCs.

We decided to address this question using the medaka gill, which works as a respiratory, sensory and osmoregulatory organ in most teleost fish. Gills are permanently exposed to circulating water and therefore have a high turnover rate (*Chrétien and Pisam, 1986*). Additionally, their growth pace must guarantee oxygen supply to meet the energetic demands of growing organismal size. Moving from the highest-level structure to the smallest, gills are organised in four pairs of branchial arches, a number which remains constant through the fish's life. Each brachial arch consists of two rows of an ever-increasing number of filaments that are added life-long at both extremes (*Figure 1A*). Primary filaments have a core from which secondary filaments, or lamellae, protrude. The lamellae are the respiratory units of the organ, and new lamellae are continually produced within each filament (*Wilson and Laurent, 2002*). Bigger fish, therefore, display more filaments that are longer than those of smaller fish, and there is a direct correlation of filament length and number and the body size of the fish (*Wilson and Laurent, 2002*).

Besides being the respiratory organ of fish, the gill has additional functions as a sensory and osmoregulatory organ (*Sundin and Nilsson, 2002*; *Wilson and Laurent, 2002*; *Jonz and Nurse, 2005*; *Hockman et al., 2017*). It contains oxygen-sensing cells (*Jonz et al., 2004*), similar to those found in the mammalian carotid body although with a different lineage history (*Hockman et al., 2017*), and mitochondrial rich cells (MRCs) (*Wilson and Laurent, 2002*) that regulate ion uptake and excretion and are identified by a distinctive Na+, K+, ATPase activity. Other cell types include pavement cells (respiratory cells of the gills), pillar cells (structural support for lamellae), globe cells (mucous secretory cells), chondrocytes (skeleton of the filaments) and vascular cells. All these cell types must be permanently produced in a coordinated manner during the post-embryonic life of

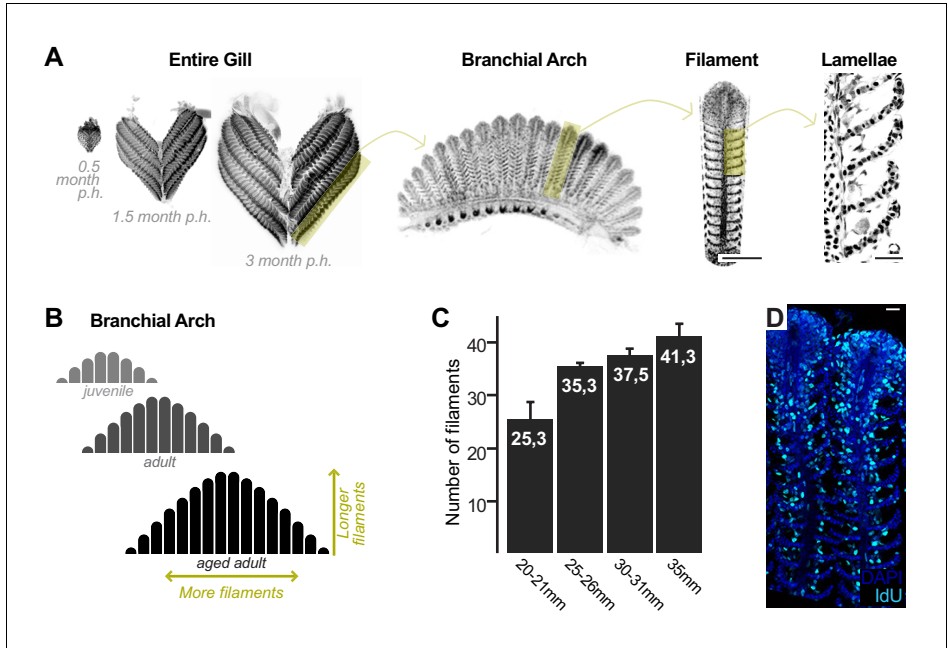

**Figure 1.** Growth and homeostasis in the medaka gill. (**A**) Enucleated entire gills of medaka at different post-embryonic times show that organ size increases during post-embryonic growth (**left**). A gill contains four pairs of branchial arches (**middle left**) that display numerous filaments (**middle right**). Filaments are composed of lamella (**right**), where gas exchange occurs. (**B**) Scheme depicting that branchial arches grow by increasing the number of filaments, and filaments grow by increasing its length. (**C**) The number of filaments per branchial arch is higher in bigger fish - x axis represents fish length, and y axis the number of filaments in the second right branchial arch. (**D**) IdU incorporation in the adult gill reflects proliferating cells all along the longitudinal axis of a filament. Scale bars are 100 μm in (**A**) filament, and 20 μm in (**A**) lamella and (**D**).

DOI: https://doi.org/10.7554/eLife.43747.002

fish. The gill constitutes, therefore, an organ that allows addressing adult stem cells during the addition and homeostatic replacement of numerous, diverse cell types.

Bona fide stem cells can only be identified and characterised by following their offspring for long periods to prove self-renewal, the defining feature of stem cells (*Clevers and Watt, 2018*). In this study, we use a lineage analysis approach that revealed growth and homeostatic stem cells in the medaka gill. We found that gill stem cells are fate-restricted, and identified at least four different lineages along each filament. By generating clones at different stages, we show that these four lineages are generated early in embryogenesis, previous to the formation of the gill. Our results also indicate that growth and homeostatic aSCs locate to different regions along the gill filaments and the branchial arches. Homeostatic stem cells have a fixed position embedded in the tissue and generate cells that move away to be integrated into an already functional unit, similarly to mammalian aSCs in the intestinal crypt (*Barker et al., 2008*). Growth stem cells, on the other hand, locate to the growing edge of filaments and are moved as filaments grow, resembling the activity of plant growth stem cells at the apical meristems (*Greb and Lohmann, 2016*). We have also found that the homeostatic aSCs can turn into growth aSCs when the apical part of a filament is ablated, revealing that the activity of a stem cell is highly plastic and depends on the local environment. Our data reveal a topological difference between growth and homeostatic stem cells, which has similar functional consequences in diverse stem cell systems.

## Results

### Medaka gills contain homeostatic and growth stem cells

The fish gill displays a significant post-embryonic expansion that reflects the activity of growth stem cells and a fast turnover rate that indicates the presence of homeostatic cells. Gills massively increment their size during medaka post-embryonic life (*Figure 1A*, left), where growth happens along two orthogonal axes. One axis represents the increase in length of each filament, and the other, the iterative addition of new filaments to a branchial arch. This way, branchial arches of an adult fish contain more filaments, which are also longer, than those of juveniles. Branchial arches in medaka continue to expand along these two axes well after sexual maturation (*Figure 1B,C*). Gills from teleost fish are exposed to the surrounding water and experience a fast turnover rate. When adult medaka fish are incubated with IdU for 48 hr, their gill filaments display a strong signal from the base to the top (*Figure 1D*), which indicates the presence of mitotically active cells all along the filament's longitudinal axis. These observations position medaka gills as an ideal system to explore the presence of growth and homeostatic stem cells within the same organ and address their similarities and differences.

### Growth stem cells locate to both growing edges of each branchial arch

We first focussed on identifying growth stem cells, by combining experimental data on clonal progression with a mathematical approach to quantify the expected behavior for stem-cell- and progenitor-mediated growth. Experimentally, clones were generated using the Gaudí toolkit, which consists of transgenic lines bearing floxed fluorescent reporter cassettes (Gaudí$^{RSG}$ or Gaudí$^{BBW2.1}$) and allows inducing either the expression or the activity of the Cre recombinase (Gaudí$^{Hsp70A.CRE}$ or Gaudí$^{Ubiq.iCRE}$, respectively). The Gaudí toolkit has already been extensively used for lineage analyses in medaka (*Centanin et al., 2014*; *Reinhardt et al., 2015*; *Lust et al., 2016*; *Aghaallaei et al., 2016*; *Seleit et al., 2017*). Clones are generated by applying subtle heat-shock treatments (when Gaudí$^{Hsp70A.CRE}$ is used) or low doses of tamoxifen (when Gaudí$^{Ubiq.iCRE}$ is used) to double transgenic animals, which results in a sparse labelling of different cells along the fish body, transmitting the label to their offspring.

The length of filaments increases from peripheral to central positions (*Figures 1A* and *2A*), regardless of the total number of filaments per branchial arch (*Leguen, 2018*). This particular arrangement suggests that the oldest and therefore longest filaments, of embryonic origin, locate to the centre of a branchial arch, while the new filaments are incorporated at the peripheral extremes either by stem cells (permanent) or progenitors (exhaustive). Conceptually, the latter two scenarios would lead to different lineage outputs. If filaments were formed from progenitor cells that are already present at the time of labelling, we would anticipate that the post-embryonic - peripheral -

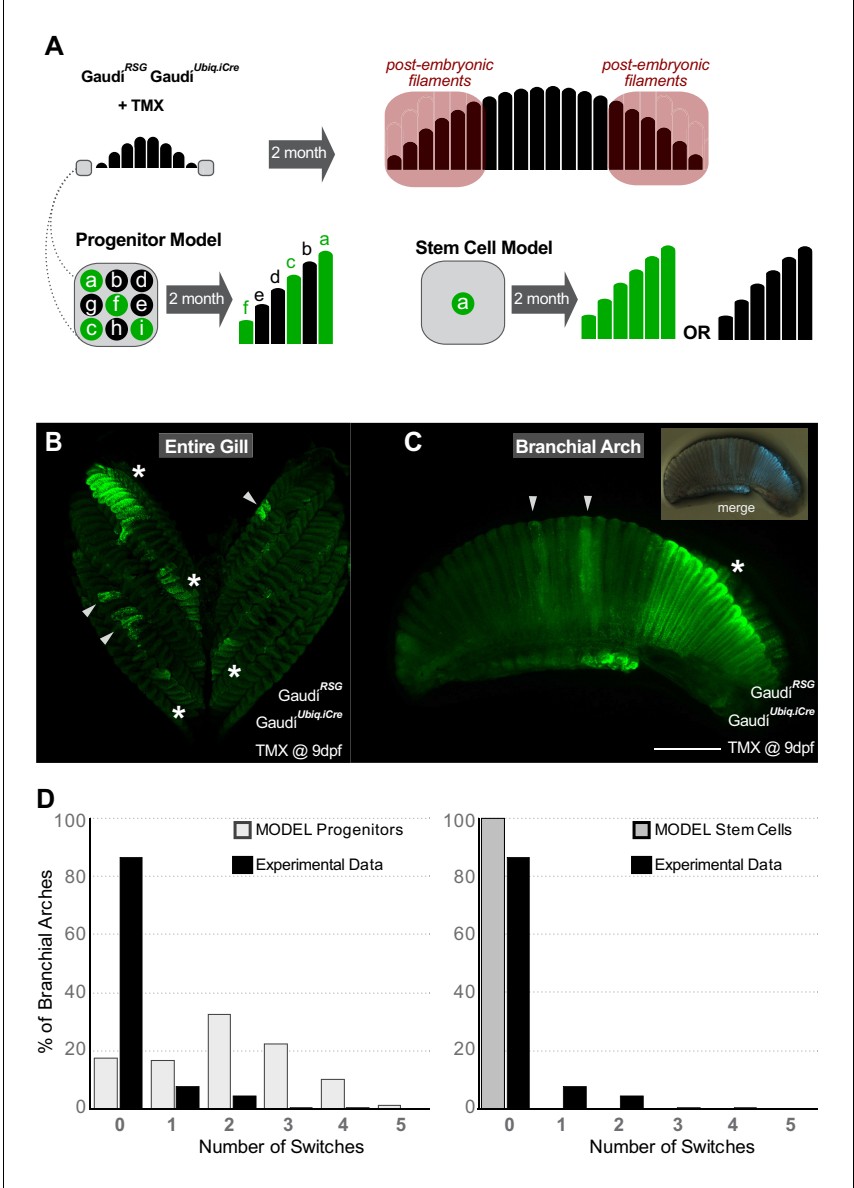

**Figure 2.** Gill stem cells located at the periphery of branchial arches generate more filaments life-long. (A) Scheme showing the expected outcome assuming a progenitor (**left bottom**) or a stem cell (**right bottom**) model. Please note that the schemes considered as 'labelled' any filament containing EGFP+ cells all along their longitudinal axis. (B) Entire gill from a double transgenic Gaudi$^{Ubiq.iCre}$ Gaudi$^{RSG}$ fish 2 month after induction with TMX. (C) Branchial arch from a double transgenic Gaudi$^{Ubiq.iCre}$ Gaudi$^{RSG}$ fish 2 months after induction with TMX. Arrowheads in B and C indicate recombined embryonic filaments located at the centre of branchial arches, and asterisks indicate stretches of peripheral filaments with the same recombination status. Note that the present resolution does not allow revealing different recombination patterns in each filament. (D) Graphs showing the distribution of switches in stretches of the six most peripheral filaments. The graphs show a comparison of the experimental data (black) to the expected distribution according to a progenitor model (light gray, left) and to a stem cell model (gray, right). Scale bar is 500 μm in (C).

DOI: https://doi.org/10.7554/eLife.43747.003

domain of adult branchial arches should contain both labelled and unlabelled filaments (*Figure 2A*, bottom left). Alternatively, if post-embryonic filaments were generated by bona fide, self-renewing stem cells, the periphery of adult branchial arches should be homogeneous in its labelling status, containing either labelled or non-labelled stretches of clonal filaments (*Figure 2A*, bottom right).

Please note, based on this reasoning, and as shown in *Figure 2A*, we define as 'labelled' any filament that contains EGFP+ cells all along the longitudinal axis; the analysis of different patterns of recombination observed are described in detail in Figure 4–8). When we analysed adult Gaudí$^{Ubiq.}$$^{iCRE}$ Gaudí$^{RSG}$ transgenic fish that had been induced for sparse recombination at old embryonic stages (nine dpf.), we observed that post-embryonic filaments at the extreme of branchial arches were grouped in either labelled or non-labelled stretches (*Figure 2B,C*, asterisks for labelled stretches and arrowheads for embryonic filaments) suggesting that they were generated by bonafide stem cells.

Our experimental data were then compared to the outcome of a computational model accounting for different scenarios for progenitor and stem cell-mediated growth. The analysis was focussed on the six most peripheral filaments of adult branchial arches (see M and M for details on filament numbers and how labelling efficiency was calculated). For each scenario, we employed stochastic simulations assigning '0' to a non-labelled filament and '1' to a labelled filament and computing the number of switches in the labelled status of two consecutive filaments, that is the number of transitions from '0-to-1' and from '1-to-0' (*Supplementary files 1* and *2*) (1000 simulations on 5000 randomly generated stretches for each experimental gill analysed, see M and M). Assuming a labelling efficiency of 50%, a progenitor-based model results in a normal distribution of switches while a stem-cell-based model shows no switches among consecutive filaments, that is contains only filaments that have a value of either 0 or 1 (see *Figure 2D* for the number of switches for each model with labelling efficiencies estimated from experiments). We have quantified both peripheral extremes of hundreds of experimental branchial arches (N > 300 6 filament stretches, N = 22 independent gills) (*Supplementary file 3*) and compared each individual branchial arch to the simulation results of the two models. For every gill analysed, the *stem cell* model explained the experimental data better than the *progenitor cell* model (*Supplementary file 4*). Altogether, our data revealed the existence of growth stem cells at the peripheral extremes of branchial arches, which generate new filaments during the post-embryonic life in medaka.

## Growth stem cells locate to the growing edge of each filament

The massive post-embryonic growth of teleost gills occurs by increasing the number but also the length of filaments. Previous data on stationary samples suggest that filaments grow from their tip (*Morgan, 1974*), and we followed two complementary dynamic approaches to characterise stem cells during filament growth. First, we exploited the high rate of cellular turnover previously observed by a pulse of IdU (*Figure 1D*), which labels mitotic cells all along the filament. We reasoned that during a chase period, cells that divide repeatedly — as expected for stem cells driving growth — would dilute their IdU content with every cell division, as previously reported for other fish tissues (*Centanin et al., 2011*). Therefore, the chase period reveals a region in the filament with a decreased signal for IdU that may, in turn, indicate where new cells are being added (*Figure 3A* illustrates the different scenarios). Indeed, all filaments analysed contained a region deprived of IdU at the most distal tip (*Figure 3B*), what stays in agreement with the previous assumptions. Complementary, we performed a clonal analysis by inducing sparse recombination using Gaudí transgenic fish. To reveal the localisation of growing clones, Gaudí$^{Ubiq.iCRE}$ Gaudí$^{RSG}$ fish were induced for recombination at 3 weeks post-fertilisation and grown for 1 month after tamoxifen treatment. We observed that clones at the proximal and middle part of the filament were small and restricted to one lamella, while the clones at the distal part contained hundreds of cells suggesting that they were generated by growth stem cells (*Figure 3C,D*).

Analysis of pulse-chase IdU experiments in entire branchial arches also suggested that the fraction of IdU-labelled cells decreased from central to peripheral filaments. While the most central filaments contain IdU-positive cells in roughly 80% of their length, filaments close to the periphery contain just few IdU cells at the basal part or even no IdU cells at all, indicating that they were produced after IdU administration. Macroscopically, IdU label had a shape of a smaller sized branchial arch nested within a non-labelled, bigger branchial arch (*Figure 3E,F*). Interestingly, while we observed that the central filaments showed a longer basal signal that becomes shorter in more peripheral filaments, the upper non-labelled fraction seemed rather stable along the central-to-periphery axis of the branchial arch (*Figure 3F*). This suggested that individual filaments had grown at comparable rates during the chase phase, highlighting the coordinated activity of the stem cells that sustained length growth in each filament. Taken together, IdU experiments revealed the growth

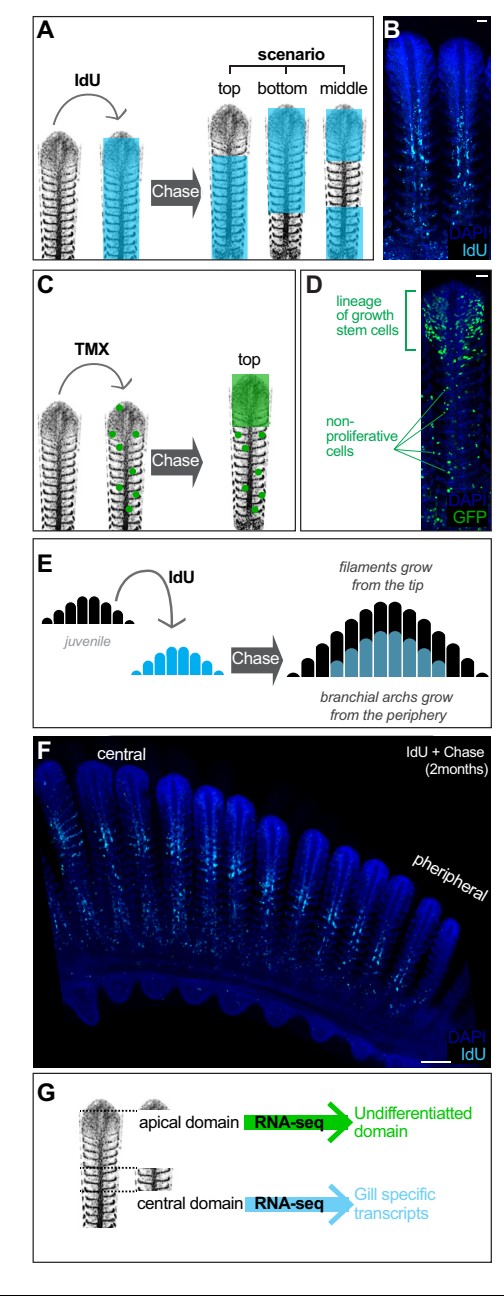

**Figure 3.** Filament growth stem cells are located at the apical tip. (**A**) Scheme showing the expected outcome of IdU *pulse and chase* experiments depending on the location of growth stem cells. (**B**) IdU *pulse and chase* experiment shows the apical region devoted of signal, indicating these cells were generated after the IdU pulse. (**C**) Scheme showing the expected outcome of a filament in which growth stem cells were labelled. (**D**) A filament from a double transgenic Gaudi$^{Ubiq.iCre}$ Gaudi$^{RSG}$ fish one month after induction with TMX shows an expanding clone in the apical region, indicating a high proliferative activity compared to clones located at other coordinates along the longitudinal axis. (**E, F**). Scheme (**E**) and data (**F**)

*Figure 3 continued on next page*

of filaments starting from their most distal extreme, and clonal analysis indicated the location of the growth stem cells at the growing tip of each filament.

We next aimed to molecularly characterise the apical, growth domain of gill filaments to complement our functional analysis using lineage tools and BrdU pulse-chase experiments. To do this, we extracted the gill from adult medaka fish, enucleated the branchial arches and manually dissected the top region of the longest filaments (*Figure 3G*, scheme). Further, the central region of gill filaments, which has been shown to contain differentiated cells in other fish species (*Laurent, 1984*; *Morgan, 1974*; *Laurent et al., 1994*), was also dissected. Total RNA from both samples were sequenced (see Materials and methods), and *Figure 3—source data 1* the differentially expressed genes were arranged according to their enrichment within the top or the central regions of the filament (*Supplementary file 5*) (*Figure 3—figure supplement 1*) (*Figure 3—source data 1*). As expected, upregulated genes in the central domain included numerous membrane transporters and channels that are characteristic of the fish gill, typically present in ionocytes and MRC cells. Indeed, GO analysis of RNA-seq data using the DAVID bioinformatic resource (https://david.ncifcrf.gov/) recognised the genes enriched in the central sample as 'Gill Tissue' (UP_TISSUE: Gill). When analysing the differential genes in the apical domain of filaments, however, the major ontology class assigned was 'Embryo' (UP_TISSUE: Embryo), which is compatible with a region enriched in undifferentiated cells like stem cells and amplifying progenitors. Although a deeper, functional analysis of the differentially expressed genes will certainly improve our understanding on the mechanistic maintenance of both domains, our molecular data and functional lineage analysis indicate the presence of a growth domain containing bona-fide stem cells at the tip of each gill filament.

## Growth stem cells are fate restricted

Gill filaments contain different cell types distributed along their longitudinal axis (*Laurent, 1984*; *Sundin and Nilsson, 2002*; *Wilson and Laurent, 2002*). Having revealed growth stem cells at the tip of each filament, we explored whether different cell types had a dedicated or a common stem cell during post-embryonic growth. Previous experiments in zebrafish on labelling cell populations at early embryonic stages revealed that

*Figure 3 continued*
showing an IdU *pulse and chase* experiment on branchial arches. The apical part of each filament and the more peripheral filaments are devoted of signal revealing the stereotypic growth of branchial arches. (**G**) Scheme showing the manual sorting of apical and middle regions for total RNA preparation. RNA-seq revealed differentiated cells in the middle region of filaments and undifferentiated cells at the apical domain. Scale Bars are 20 μm in (**B**) and (**D**), and 100 μm in (**F**).
DOI: https://doi.org/10.7554/eLife.43747.004
The following source data and figure supplement are available for figure 3:

**Source data 1.** Transcriptome of apical and medial domains in a gill filament.
DOI: https://doi.org/10.7554/eLife.43747.006
**Figure supplement 1.** Technical analysis of the RNA-seq data.
DOI: https://doi.org/10.7554/eLife.43747.005

neuroendocrine cells (NECs) are derived from the endoderm (*Hockman et al., 2017*), while pillar cells have a neural crest origin (*Mongera et al., 2013*). We followed a holistic approach to address the potency of gill stem cells once the organ is formed, by using inducible ubiquitous drivers to potentially label all possible lineages within a gill filament. We induced sparse recombination at 8 dpf. in Gaudí$^{Ubiq.iCRE}$ Gaudí$^{RSG}$ double transgenic fish and grew them to adulthood. We selected gills with a reduced number of EGFP-positive clones (*Figure 4A*) and imaged branchial arches and gill filaments with cellular resolution (*Figure 4B–F*). Our analysis revealed the presence of four different recombination patterns illustrating the lineage of different types of growth stem cells (*Figure 4C–F*, patterns 1 to 4). Moreover, this lineage analysis approach showed that growth stem cells at the tip of gill filaments are indeed fate-restricted, and hence, the most apical domain of a filament hosts different growth stem cells with complementary potential.

Noticeable, recombined filaments displayed the same lineage patterns spanning from their base, that is juvenile domain, to their tip, that is adult domain, (*Figure 4C–F*) (N > 200 recombined filaments) indicating that growth stem cells maintain both their activity and their potency during a lifetime. A detailed description of the different cell types included in each lineage largely exceeds the scope of this study. Broadly speaking, labelled cells in pattern 1 (*Figure 4C,G–H*) are epithelial cells covering the lamellae and the interlamellar space, including MRC cells as revealed by expression of the Na+/K + ATPase (*Figure 4H*). Pattern 3 and 4 display a reduced number of labelled cells, sparsely distributed along the filament (pattern 3) or surrounding the gill ray (pattern 4) (*Video 1* and *Video 2*, respectively). Pattern 2 consists of labelled pillar cells and chondrocytes of the gill ray (*Figure 4I,I'*, and reconstructions in *Video 3*), both easily distinguishable by their location and unique nuclear morphology. Both cell types were previously reported as neural crest derivatives (*Mongera et al., 2013*), and our results demonstrate that they are produced by a common stem cell in every filament during the post-embryonic growth of medaka.

We revealed in the previous sections that growth stem cells at the periphery of branchial arches (*br-arch*SCs) generate new filaments, and we showed that each filament contains, in turn, growth stem cells (*filam*SCs) of different fates. To address whether the fate of *filam*SCs is acquired when filaments are formed or set up already in *br-arch*SCs and maintained life-long, we exploited the stretches of labelled — and therefore clonal — filaments observed at the periphery of branchial arches in adult Gaudí$^{RSG}$ Gaudí$^{Ubiq.iCRE}$ fish induced for recombination during late embryogenesis (*Figures 2B* and *4A,B*). We reasoned that if a labelled *br-arch*SC is fate-restricted, the consecutive filaments formed from it should display an identical recombination pattern since *filam*SCs would have inherited the same fate-restriction from their common *br-arch*SC. Alternatively, if *filam*SCs would acquire the fate-restriction when each filament is formed, then a stretch of clonal filaments should display different recombination patterns, based on the independent fate acquisition at the onset of filament formation (schemes in *Figure 5A*). We have focussed on 153 branchial arch extremes that started with a labelled filament (N = 83 for *rec. pattern 1*, N = 44 for *rec. pattern 2*, N = 22 for *rec. pattern three* and N = 4 for *rec. pattern 4*), and 97.4% were followed by a filament with the same recombination pattern (*Figure 5B–E*) (*Supplementary file 6*). Moreover, 81.7% of stretches maintained the same recombination pattern for six or more filaments, indicating that the labelled cell-of-origin for post-embryonic filaments was already fate-restricted. Altogether, our data revealed that a branchial arch contains fate-restricted growth *br-arch*SCs at its peripheral extremes that produce growth *filam*SCs stem cells with the same fate-restriction.

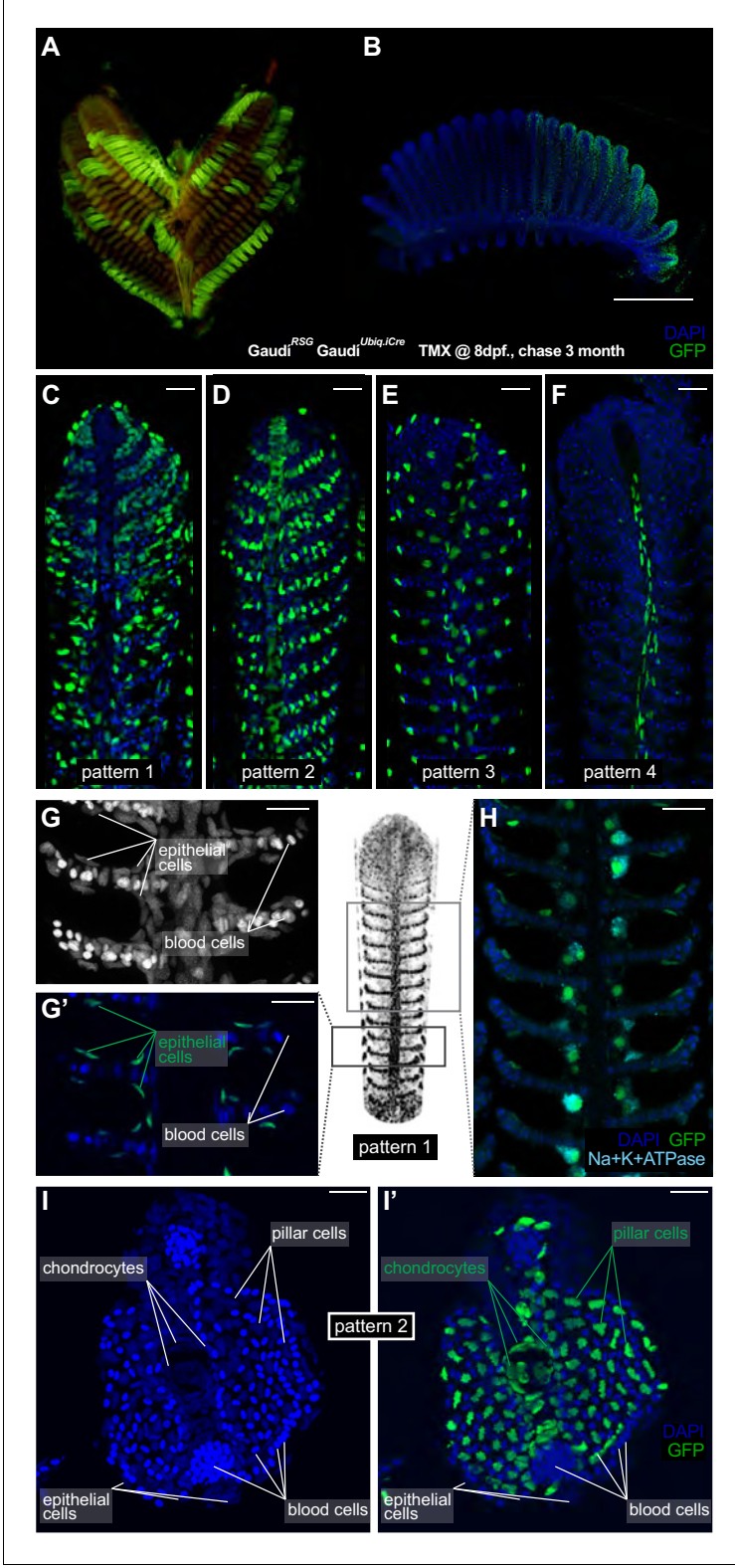

**Figure 4.** Filament growth stem cells are fate restricted. (**A–B**) A gill (**A**) and a branchial arch (**B**) from a double transgenic Gaudi$^{Ubiq.iCre}$ Gaudi$^{RSG}$ fish two month after induction with TMX. (**C–F**) Confocal images from filaments in A, B, stained for EGFP and DAPI to reveal the cellular composition of different clones. Four different recombination patterns were identified. (**G, G'**) A detailed view of pattern 1(C) shows recombined epithelial cells covering each lamella. (**H**) Co-staining with an anti-Na$^+$K$^+$ATP-ase antibody confirms that MRC cells are clonal to

*Figure 4 continued on next page*

*Figure 4 continued*

other epithelial cells in the filament. (I, I') Cross-section of a filament that displays pattern 2 (D). DAPI staining allows identifying blood cells (strong signal, small round nuclei), pillar cells (weaker signal, star-shaped nuclei), and chondrocytes (elongated nuclei at the central core of the filament) (I). The lineage tracker EGFP reveals that chondrocytes and pillar cells are clonal along a filament (I'). Scale Bars are 500 μm in (B), 20 μm in (C–H) and (I).
DOI: https://doi.org/10.7554/eLife.43747.007

Our results revealed that post-embryonic gill growth depends on the coordinated activity of at least four stem cells, which maintain different, non-overlapping lineages. Do these different stem cells share a common embryonic progenitor? To address this, we induced recombination of Gaudí$^{RSG}$ Gaudí$^{Hsp70:CRE}$ embryos at early, mid and late embryonic stages (stg 20, 26, 34 and 39, corresponding to 2 dpf to 9 dpf in medaka) and grew them for 3 months. Confocal analysis of these samples (*Figure 5—figure supplement 1A,B*) revealed the same scenario as previously reported, with four different fate-restricted stem cells (pattern 1 N = 87 filaments, pattern 2 N = 62 filaments, pattern 3 N = 86 filaments, and pattern 4 N = 49 filaments; N = 34 branchial arches). We have observed cases in which the same recombination pattern is displayed along the entire branchial arch (*Figure 5—figure supplement 1B*)(pattern 1 N = 1 branchial arch, pattern 3 N = 2 branchial arches, pattern 4 N = 1 branchial arch), indicating that fate-restriction had already occurred when brachial arches were formed. The earliest experimental approach for lineage analysis in fish involves transplanting blastomeres from a line containing a ubiquitous genetic label into a non-labelled host blastula. When we performed transplantations using Gaudí$^{LoxPOUT}$ or Gaudí$^{BBW}$ as labelled donors (*Centanin et al., 2014*) and grew fish for 3 months, clones in the gills displayed the same patterns previously described (*Figure 5—figure supplement 1C–C'*, *Figure 6*). Therefore, our lineage analysis involving blastula transplantation, heat-shock-induced recombination during early embryonic stages and tamoxifen-induced recombination during late embryonic stages indicated the existence of fate-restricted gill stem cells that have independent embryonic origins.

## Wild type - *p53$^{-/-}$* chimeras reveal functional differences among lineages

While some of the most studied adult stem cells in vertebrates are multipotent (*Clevers and Watt, 2018*), our data suggest the fish gill is maintained by the synchronised activity of at least four stem cells of different fates. This growth feature of the medaka gill permits studies of the relative contribution of each lineage to the overall growth of filaments and the differential effect of mutations on the diverse stem cell types. We created chimeras mixing wild type and *p53$^{E241X}$* mutant cells at blastula stages in an attempt to generate lineage-specific mutant clones. P53 has first been reported as a tumour suppressor and is implicated in growth coordination and self-renewal of adult stem cells in different systems (*Jain and Barton, 2018*; *Mesquita et al., 2010*; *Pearson and Sánchez Alvarado, 2010*). The *p53$^{E241X}$* medaka mutant is viable and fertile (*Taniguchi et al., 2006*), and adult *p53$^{E241X}$* display branchial arches that have the same morphology and a comparable size than wild-type branchial arches in medaka (*Figure 6—figure supplement 1*). Therefore, we proceeded to generate chimeras by transplanting EGFP labelled *p53$^{E241X}$* mutant blastocysts into a wild type blastula (Gaudí$^{LoxPOUT}$ *p53$^{E241X}$* to WT), and vice-versa, transplanting EGFP labelled wild-type blastocysts into a *p53$^{E241X}$* mutant blastula (Gaudí$^{LoxPOUT}$ to *p53$^{E241X}$*). We selected chimeras that displayed clearly visible EGFP clones by the end of embryogenesis, which were isolated and maintained for at least 3 months. In both cases, we

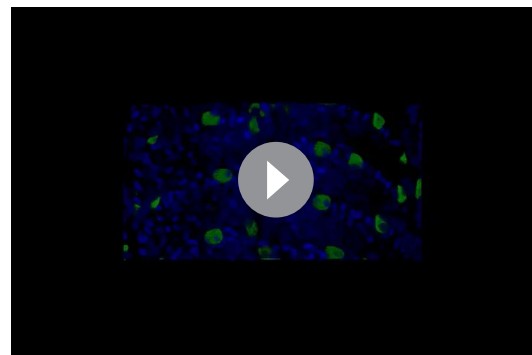

**Video 1.** 3D reconstruction of a pattern 3-labelled filament. A middle section of filament in an adult Gaudí$^{Ubiq.iCre}$ Gaudí$^{RSG}$ fish that was induced for recombination at late embryonic stages. The filament shows the lineage of a growth stem cell that labels pattern 3.
DOI: https://doi.org/10.7554/eLife.43747.008

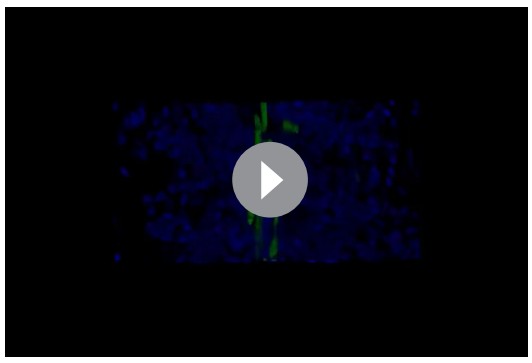 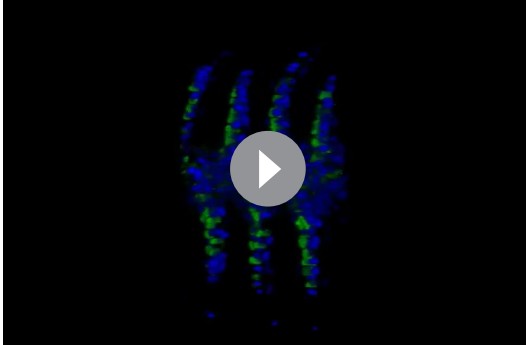

**Video 2.** 3D reconstruction of a pattern 4-labelled filament. A middle section of filament in an adult Gaudi^Ubiq.iCre GaudíRSG fish that was induced for recombination at late embryonic stages. The filament shows the lineage of a growth stem cell that labels pattern 4.
DOI: https://doi.org/10.7554/eLife.43747.009

**Video 3.** 3D reconstruction of a pattern 2-labelled filament. A middle section of filament in an adult Gaudi^Ubiq.iCre GaudíRSG fish that was induced for recombination at late embryonic stages. The filament shows the lineage of a growth stem cell that labels pattern 2.
DOI: https://doi.org/10.7554/eLife.43747.010

obtained branchial arches displaying filaments labelled in one pattern only (*Figure 6A,B*), which indicates that P53 is not critical to maintain fate-restriction in gill stem cells.

The generation of *p53*^-/-- to-WT chimeras allowed us to create composite filaments with an EGFP⁺ labelled mutant lineage and three non-labelled lineages. We focussed on filaments labelled in pattern 2 (*Figure 6A*) (N = 4 filaments in two branchial arches) and observed that in both length and morphology chimeras were indistinguishable from neighbour wild-type filaments in the same branchial arch. This trend was consistent in *p53*^+/-- to-WT chimeras (pattern 2 N = 2 filaments, pattern 4 N = 7 filaments). Notably, however, in branchial arches of WT-to-*p53*^-/- chimeras, composite filaments were clearly shorter than their wild type neighbours (*Figure 6B*). Shorter filaments were filaments in which pattern 2 came from donor cells and the rest of the lineages were from the host (*Figure 6B'–B'''*) (19/19 filaments labelled in pattern 2 were shorter than their immediate neighbours). Composite filaments in which pattern 3 came from donor cells displayed a regular size (*Figure 6B'–B'''*) (58/58 filaments labelled in pattern 3 were indistinguishable from their neighbours). Thus, our data from WT-to-*p53*^-/- chimeras reveal that manipulating just one lineage in the organ results in a massive growth phenotype. Critically, as these growth defects are limited to the filaments labelled in pattern 2 these data suggest that the different lineages may have unequal roles during filament growth. It will be important, therefore, to define the molecular mechanisms downstream of P53 that are responsible for sustaining proper growth in future studies.

## Homeostatic stem cells locate to the base of each lamella

Branchial arches grow during post-embryonic life by adding more filaments (*Figure 1A–C*), and filaments grow in length by adding more lamellae (*Figure 1A*, *Figure 7A*). Noticeable, the length of consecutive lamellae does not increase with time along a filament (*Figure 7B*), resulting in basal and apical lamellae having comparable sizes (basal: 35,72 ± 1,93 um and apical: 34,38 ± 4,04 um N = 6 lamellae of each). This also holds true when comparing the length of lamellae from long (central, embryonic) and short (peripheral, post-embryonic) filaments, and comparing lamellae from medaka of different body length. Lamellae, therefore, maintain their size despite containing proliferative cells (*Laurent, 1984*; *Laurent et al., 1994*), a scenario that resembles most mammalian stem cell systems in adults, such as the intestinal crypt or the hair follicle. Previous studies have reported mitotic figures along with the filament core in histological sections of various teleost fish. To address the presence and location of proliferating cells in the lamellae of medaka, we performed shorter IdU pulses (12 hr) and observed that most lamellae contained positive cells at the proximal extreme (*Figure 7C*), adjacent to the central blood vessels and the gill ray.

We next performed a lineage analysis of gill stem cells during homeostasis, focussing on the lamellae since they constitute naturallyoccurring physical compartments that facilitate the analysis of

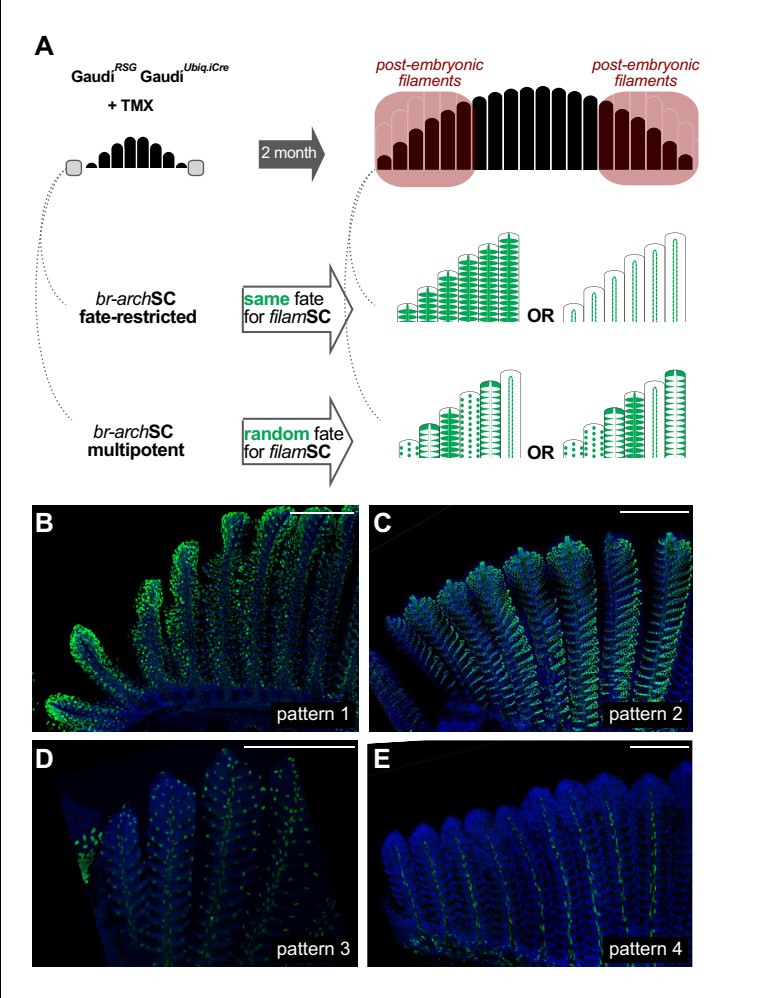

**Figure 5.** Branchial arch stem cells are fate restricted. (**A**) Scheme showing the expected outcome assuming that *br-arch*SCs are fate restricted (**middle**) or multi-potent (**bottom**). The recombination pattern of consecutive filaments would be identical if generated by fate restricted *br-arch*SCs, and non-identical if derived from a multipotent *br-arch*SC. (**B–E**) Confocal images show an identical recombination pattern in consecutive peripheral filaments for pattern 1 (**B**), pattern 2 (**C**), pattern 3 (**D**) and pattern 4 (**E**). Scale Bars are 200 µm in (**B**).
DOI: https://doi.org/10.7554/eLife.43747.011

The following figure supplement is available for figure 5:

**Figure supplement 1.** Early embryonic recombination and transplantations at blastula stage indicate fate-restricted stem cells in the fish gill.
DOI: https://doi.org/10.7554/eLife.43747.012

clonal progression. We used double transgenic Gaudí*Ubiq.iCre* Gaudí*RSG* adults that were grown for 3 additional weeks after clonal labelling, and focussed on those containing only a few recombined lamellae per branchial arch (labelling efficiency less than 0.5%). Detailed analysis of lamellae located far away from the filaments' growing tip revealed clones of labelled cells spanning from the proximal to the distal extreme of the lamella (*Figure 7D,E*). The clones ranged from a few pillar cells (*Figure 7D,D'*) to most pillar cells in the lamella (*Figure 7D'', E* and *Video 4*). This dataset reflects the activity of stem cells contributing to a structure that does not increase in size but renews the cells within — that is homeostatic stem cells. Our results, therefore, indicate the presence of homeostatic pillar stem cells at the base of each lamella in medaka gills.

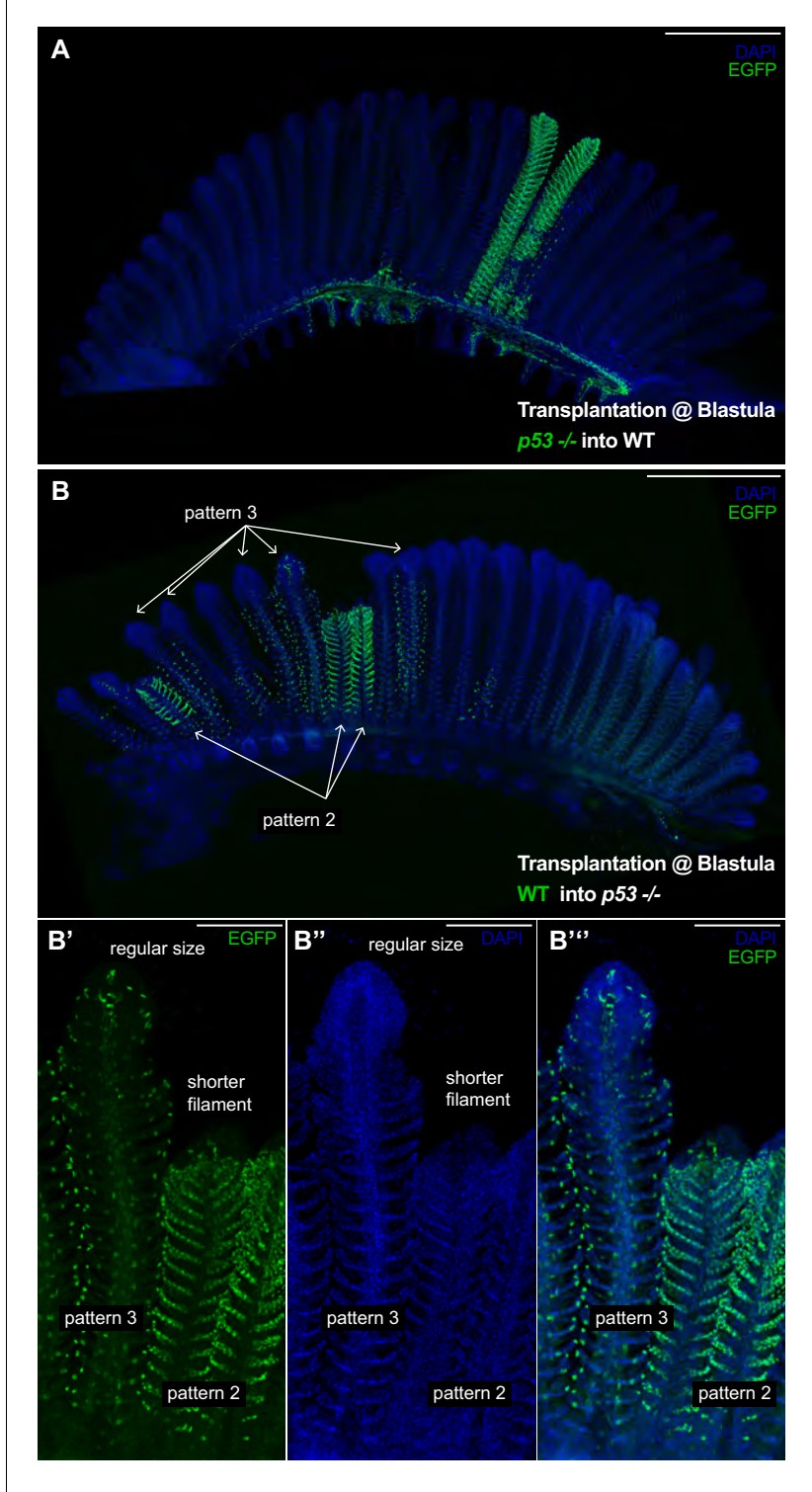

**Figure 6.** *p53* Coordinates growth stem cells in a lineage-specific manner. (**A**) Branchial arch of a *p53*<sup>-/-</sup> -to-WT chimera, where *p53*<sup>-/-</sup> mutant cells are labelled in green. Composite filaments display the proper length as compared to non-labelled neighbour filaments. (**B**) Branchial arch of a WT-to-*p53*<sup>-/-</sup> chimera, where WT cells are labelled in green. Composite filaments are shorter than their neighbours only when pattern 2 comes from the donor - Note the short size of the right filament in B'-B'' compared to the filament at the left. Scale Bars are 500 µm in (**A, B**) and 100 µm in (**B'–B'''**).

DOI: https://doi.org/10.7554/eLife.43747.013

*Figure 6 continued on next page*

*Figure 6 continued*

The following figure supplement is available for figure 6:

**Figure supplement 1.** Branchial Arch of *p53^E241X* mutant.

DOI: https://doi.org/10.7554/eLife.43747.014

## The homeostatic domain can restore filament growth

Our lineage analysis revealed distinct locations for both growth and homeostatic stem cells along gill filaments. The growth domain of filaments is always at the top, while the homeostatic domain extends along the longitudinal axis (*Figure 8A*). Our lineage analysis also revealed that growth and homeostatic stem cells are clonal since all homeostatic stem cells within a lineage are labelled when a filament has the corresponding labelled growth *filam*SC (*Figure 4C–F*). We then wondered about their different behaviour; while growth stem cells are displaced by the progeny they generate, homeostatic stem cells maintain their position while pushing their progeny away. These different locations along the filament might constitute dissimilar physical niches. It has indeed been shown in other teleost fish that the growing edge where growth stem cells host is subjected to less spatial restriction than the gill ray niche (*Morgan, 1974*). On the other hand, there is a strong extracellular

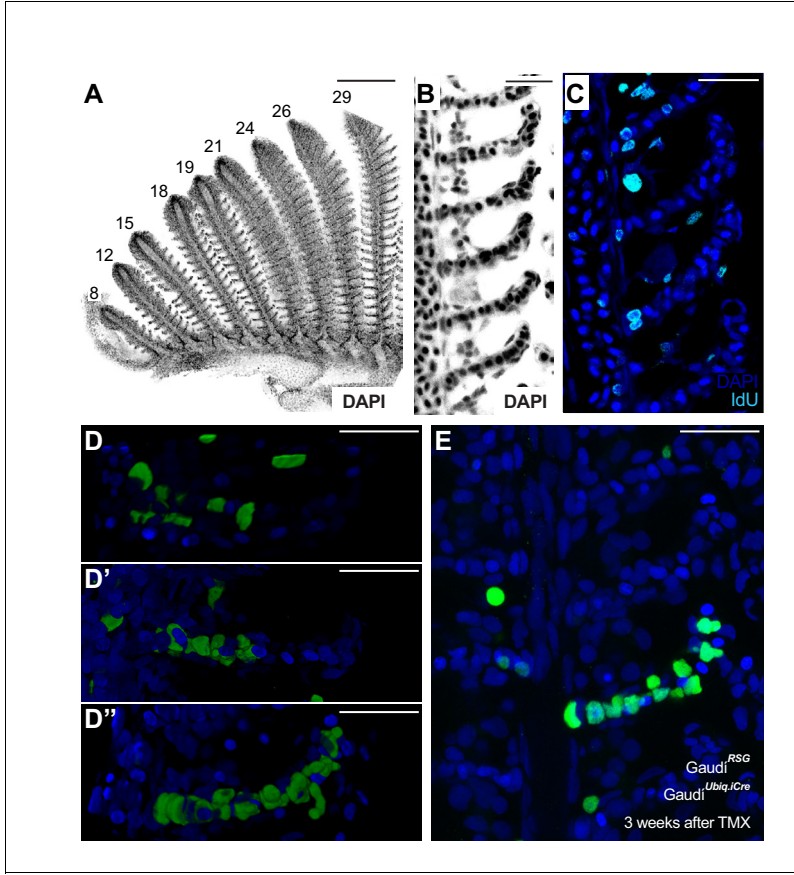

**Figure 7.** Homeostatic stem cells locate to the base of each lamella. (**A**) DAPI image of peripheral filaments indicating the increasing number of lamellae per filament. (**B**) DAPI image of consecutive lamellae along a filament reveals that lamellae do not increase their size. (**C**) IdU pulse reveals proliferative cells at the base of the lamellae. (**D–E**) EGFP cells indicating clonal progression of clones in double transgenic Gaudí^Ubiq.iCre^ Gaudí^RSG^ fish 1 month after induction with TMX during adulthood. Clones of pillar cells progress from the base to the distal part of a lamellae (**D''**, **E**). Scale Bars are 200 µm in (**A**), and 20 µm in (**B–E**).
DOI: https://doi.org/10.7554/eLife.43747.015

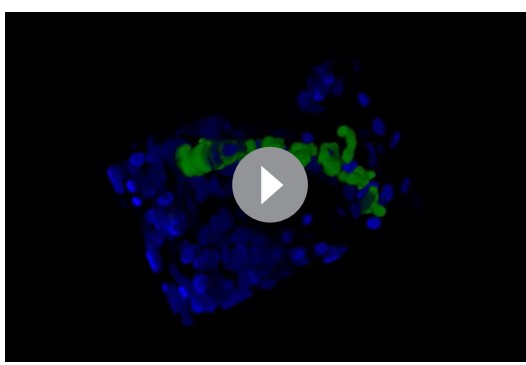

**Video 4.** 3D reconstruction of a pattern 2-labelled lamella. A middle section of filament in an adult Gaudí^{Ubiq.iCre} Gaudí^{RSG} fish that was induced for recombination at late embryonic stages. The filament shows the lineage of a homeostatic stem cell that labels pattern 2 only in one lamella.
DOI: https://doi.org/10.7554/eLife.43747.016

matrix rich in collagen and secreted mainly by chondrocytes and early pillar cells across the filament (*Morgan, 1974*), adjacent to the place in which we characterised homeostatic stem cells.

We speculated that modifying the close environment of homeostatic stem cells by ablating the growing zone of a filament could elicit a growth response from the homeostatic domain. We, therefore, ablated a reduced number of filaments (3 to 4) by removing their upper region using surgery scissors (*Figure 8A*). Filaments are densely packed along a branchial arch, but they conform detached units all along their longitudinal axis, which allowed removing the growth domain of 3 to 4 filaments and using the neighbour filaments in the same branchial arch as internal controls. When experimental fish were grown for a month after ablation, we could still recognise the ablated filaments due to their shorter length, compared to that of their neighbour, non-ablated filaments (*Figure 8B*). Ablated filaments, however, restored the characteristic morphology of a growth domain at their most upper extreme (*Figure 8C,D*). Additionally, IdU incorporation showed that the *new* growth domains were proliferative, showing a similar IdU label than non-ablated filaments in the same branchial arch (*Figure 8E*). Our lineage analysis during homeostatic growth in medaka revealed different growth and homeostatic stem cells in each filament that maintained their fate during the entire life of the fish. We, therefore, wanted to assess whether the reconstitution of a filament growth domain after injury required cells from all different lineages or if alternatively, cells from a given lineage would change their fate to contribute to multiple recombination patterns (*Figure 4C–F*). Injury paradigms have been shown to affect the fate commitment of stem cells in different models (*Van Keymeulen et al., 2011*; *Suetsugu-Maki et al., 2012*) while in others, proliferative cells maintain their fate during the regeneration process (*Kragl et al., 2009*; *Knopf et al., 2011*).

To address the nature of the cells re-establishing the growth domain, the same injury assay was performed on Gaudí^{Ubiq.iCRE} Gaudí^{RSG} transgenic fish that had been induced for sparse recombination at late embryonic stage (eight dpf) and grown for two months. When we analysed these samples 3 weeks after injury, we observed that the recombination pattern of the basal, non-injured region was identical to the recombination pattern of the newly generated zone (*Figure 8F–I*) (N = 30 filaments in six branchial arches, N = 17 for pattern 1, N = 11 for pattern 2, N = 2 for pattern 3). These results indicate that the re-established growth zone is formed by an ensemble of cells from the different lineages, and strongly suggest that homeostatic stem cells within all lineages can be converted to growth stem cells during regeneration. Our data definitively reveal that filaments possess the ability to resume growth from the homeostatic domain in a process that requires cells from different lineages. Overall, we propose from our observations that the different niches – physical and/or molecular - along the filament could operate as main regulators of the homeostatic-or-growth activity for stem cells in the fish gill.

## Discussion

In this study, we use genetic lineage analysis and mathematical modelling to reveal the rationale behind the permanent post-embryonic growth in a vertebrate organ. We introduce the fish gill, and particularly branchial arches, as a new model system that displays an exquisite temporal/spatial organisation, and use it to characterise growth and homeostatic stem cells. We reveal two domains harbouring growth stem cells: both extremes of each branchial arch contain *br-arch*SCs, which in turn generate *filam*SCs that locate to the tip of newly formed filaments. Additionally, *filam*SCs generate homeostatic stem cells at the lamellae along the longitudinal axis of the filament. The peripheral-to-central axis of branchial arches reflects a young-to-old filament order, and the longitudinal axis of a

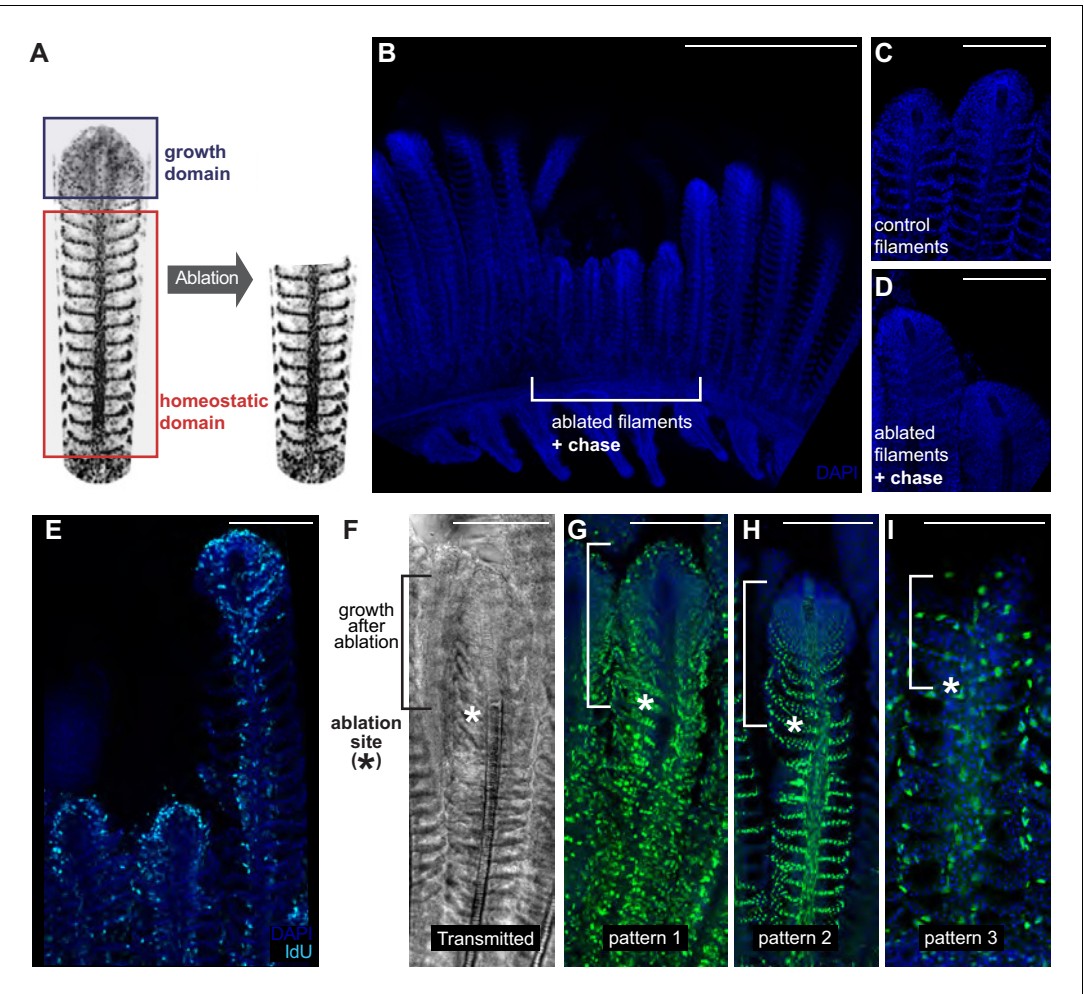

**Figure 8.** The homeostatic domain sustains growth after filament ablation. (**A**) Scheme of the ablation procedure. The growth domain and the upper part of the homeostatic domain are mechanically ablated. (**B**) DAPI image of control filaments shows an intact growth domain at the top. (**C**) DAPI image of injured filaments after a chase of 1 month shows a regenerated growth domain. (**D**) During the duration of the experiment, ablated filaments were unable to reach the length of their neighbour, non-ablated filaments. (**E**) IdU-positive cells are detected at the growth domain of both intact and ablated treatment. (**F–I**) Transmitted (**F**) and fluorescent (**G–I**) images of gill filaments from recombined Gaudi*Ubiq.iCre* Gaudi*RSG* fish, 3 weeks post-ablation. The upper fraction containing the growth domain was generated after the ablation (highlighted in F). The recombination pattern observed in the regenerated zone is identical to the recombination pattern that the filaments had before ablation, indicating that cells maintain their fate during the regenerative response. Scale Bars are 500 µm in (**A**) and 100 µm in (**C–I**).
DOI: https://doi.org/10.7554/eLife.43747.017

filament reflects a young-to-old lamellae order. The two growth stem cells and the one homeostatic stem cell types are clonal and organised in a hierarchical manner.

Our observations indicate that the relative position within the organ has a major impact on the growth vs homeostatic activity of stem cells. We have found that when the growth domain of a filament is lost, the homeostatic domain is able to generate a new, functional growth domain. This observation suggests that physical or molecular modifications in the local environment (relaxation of the inner core, or the absence of a repressive signal, respectively) could convert homeostatic stem cells into growth stem cells. In the absence of specific markers to label homeostatic stem cells before the ablation, however, we cannot discard the presence of quiescent stem cells that get activated after injury, nor the possibility of injury-triggered trans-differentiation as shown in the zebrafish caudal fin (*Knopf et al., 2011*).

Permanent post-embryonic growth is a challenging feature for an organism, since new cells have to be incorporated into a functional organ without affecting its physiological activity. Restricting growth stem cells to the growing edge is an effective way to compartmentalise cell addition and organ function. Strikingly, the location of growth stem cells in gill filaments is highly reminiscent of the overall topology of meristems in plants (*Greb and Lohmann, 2016*). In both systems, axis extension occurs by the sustained activity of stem cells that locate to the growing edge. These stem cells consistently remain at the growing zone, while their progeny start differentiation programs and occupy a final location at the coordinates in which they were born. It is to note that other ever-growing organs in fish follow the same growing principle, with tissue stem cells located at the growing edge and differentiated progeny left behind, as it has been nicely shown for different cell types in the zebrafish caudal fin (*Tu and Johnson, 2011*) and the medaka neural retina and retinal epithelium (*Centanin et al., 2011*; *Centanin et al., 2014*). Since stem cells are thought to have evolved independently in the vegetal and the animal lineages (*Meyerowitz, 2002*; *Scheres, 2007*), our results illustrate how the same rationale to sustain permanent growth can be adopted in the most diverse systems.

We have performed an organ-scale lineage analysis at cellular resolution and found that growth stem cells and homeostatic stem cells are fate-restricted. We used diverse un-biased labelling approaches (sub-optimal tamoxifen concentrations using an ubiquitously-expressed inducible [ErT2]-CRE, heat-shock-induced expression of CRE, transplantation of few permanently-labelled cells at the blastula stage) to identify at least four different fate-restricted gill stem cells, which generate reproducible labelling patterns along gill filaments. Our experiments indicate that these stem cells do not share a common early progenitor, but rather have independent embryonic origins. Since each filament contains all four fate-restricted stem cells (we have not observed filaments lacking one entire lineage), our results determine that the growth zone of a gill filament is indeed an *ensemble* — a group of stem cells with different potencies that work in an interconnected manner. Two relevant avenues open from this analysis, namely: a) how stem cells are recruited together to a newly forming filament, a process that happens hundreds of times during the lifetime of a medaka fish and thousands of times in longer-lived teleost fish, and b) how stem cells coordinate their activity to maintain the ratio of cell types in the individual filaments to guarantee its proper growth. One fundamental aspect to start addressing coordination is to define the number of stem cells for each lineage, a parameter that proved to be hard to estimate for most vertebrate organs. The prediction for gill filaments is that they contained a very reduced number of stem cells, for they generate all-or-none labelled filaments of a given cell type reflecting a clonal nature.

We exploited the fate-restriction of growth stem cells in the gill system by generating chimeras that contain wild-type and a *p53* mutant cells, and focused on filaments that display one lineage coming from donor cells and the other three coming from host cells. Our experiments revealed that P53 is indeed necessary to establish/maintain the coordinated growth of neighbour filaments. Strikingly, we observed growth phenotypes only when the lineage of pillar cells was involved: *p53* mutant filaments that contain wild-type pillar cells can not growth at the same pace than fully *p53* mutant filaments in the same branchial arch. Wild-type cells in other lineages did not produce the same phenotype, which we interpret as a differential hierarchical organisation of the different fate-restricted stem cells sustaining filament growth. Additionally, our experiments working with genetic chimeras and also the ones in which we ablate the growth domain of wild type filaments show that filament growth is a autonomous - that is a filament's growth rate does not adjust to the size of its neighbour filaments. This observation is particularly intriguing considering the order in filament size along a branchial arch, where every filament is larger that its younger neighbour and shorter than its older neighbour. Altogether, our results position the fish gill as an ideal system to quantitatively explore stem cell niches hosting multiple lineage-restricted stem cells, particularly the growth coordination within and between functional units.

In most adult mammalian organs, stem cells maintain homeostasis by generating new cells that will replace those lost during physiological or pathological conditions. We have functionally identified homeostatic stem cells in the fish gill, and focussed on the ones generating pillar cells. Our lineage analysis demonstrates that growth and homeostatic stem cells are clonal along a filament, where the former generate the latter. The most obvious difference between these two stem cell types is their relative position; growth stem cells are located at the growing tip, beyond the rigid core that physically sustains the structure of the filament, while homeostatic stem cells are

embedded inside the tissue, adjacent to the collagen-rich chondrocyte column. It is to note that both the function and the relative location of the gill homeostatic stem cells match those of the mammalian homeostatic stem cells, being located at a fixed position and displacing their progeny far away - as it is observed for intestinal stem cells, skin stem cells and oesophagus stem cells among others (*Barker et al., 2008*; *Blanpain and Fuchs, 2009*; *Seery, 2002*). The functional comparison of growth and homeostatic stem cells in the gill suggests the existence of a physical niche that would restrict stem cells to their homeostatic role, preventing them to drive growth. Indeed, our RNA-seq data comparing the transcriptomes from the growth domain and the central domains revealed numerous Integrins and membrane proteins overrepresented at the growing edge, supporting the notion of a different physical niche mediating filament growth.

We believe that during vertebrate evolution, the transition from lower (ever-growing) to higher (size-fixed) vertebrates involved restraining the growth activity of adult stem cells. One of the main functions of mammalian physical niches, in this view, would be to restrict stem cells to their homeostatic function. Many stem-cell-related pathological conditions in mammals involve changes in the microenvironment including physical aspects of the niche (*Brabletz et al., 2001*; *Vermeulen et al., 2010*; *Ye et al., 2015*; *Oskarsson et al., 2011*; *Liu et al., 2012*; *Butcher et al., 2009*), suggesting that homeostatic stem cells could drive growth in that context. Along the same line, the extensive work using organoids that are generated from adult homeostatic stem cells, like intestinal stem cells, (*Sato et al., 2009*; *Kretzschmar and Clevers, 2016*), demonstrates that healthy aSCs have indeed the capacity to drive growth under experimental conditions and when removed from their physiological niche. Our work, therefore, illustrates how different niches affect the functional output of clonal stem cells driving growth and homeostatic replacement in an intact in vivo model.

## Materials and methods

**Key resources table**

| Reagent type (species) or resource | Designation | Source or reference | Identifiers | Additional information |
|---|---|---|---|---|
| Strain, strain background (*Oryzias latipes*) | Cab | | | wild type stock, southern population |
| Genetic reagent (*O. latipes*) | p53 E241X mutant | *Taniguchi et al., 2006* | | |
| Genetic reagent (*O. latipes*) | GaudíUbiq.iCre | *Centanin et al., 2014* | | |
| Genetic reagent (*O. latipes*) | GaudíHsp70.A | *Centanin et al., 2014* | | |
| Genetic reagent (*O. latipes*) | GaudíRSG | *Centanin et al., 2014* | | |
| Genetic reagent (*O. latipes*) | GaudiLoxPOUT | *Centanin et al., 2014* | | |
| Genetic reagent (*O. latipes*) | GaudiBBW | *Centanin et al., 2014* | | |
| Antibody | a-EGFP (Rabbit IgG polyclonal) | Invitrogen (now Thermo Fischer) | CAB4211; RRID: AB_10709851 | Dilution 1:750 |
| Antibody | a-EGFP (Chicken IgY polyclonal) | life technologies | A10262; RRID: AB_2534023 | Dilution 1:750 |
| Antibody | a-Na + K + ATP-ase (Rabbit monoclonal) | Abcam | ab76020, EP1845Y | Dilution 1:200 |
| Antibody | a-BrdU/IdU (Mouse IgG monoclonal) | Becton Dickinson | 347580 | Dilution 1:50 |
| Antibody | Alexa 488 Goat a-Rabbit | Invitrogen (now Thermo Fischer) | A-11034 | Dilution 1:500 |

*Continued on next page*

*Continued*

| Reagent type (species) or resource | Designation | Source or reference | Identifiers | Additional information |
|---|---|---|---|---|
| Antibody | Alexa 488 Donkey a-Chicken | Invitrogen (now Thermo Fischer) | 703-545-155 | Dilution 1:500 |
| Antibody | Alexa 647 Goat a-Rabbit | Life Technologies | A-21245 | Dilution 1:500 |
| Antibody | Alexa 647 Goat a-Rabbit | Life Technologies | A-21245 | Dilution 1:500 |
| Antibody | Cy5 Donkey a-Mouse | Jackson | 715-175-151 | Dilution 1:500 |
| Chemical compound, drug | tamoxifen | Sigma | T5648 | |
| Chemical compound, drug | tricaine | Sigma-Aldrich | A5040-25G | |
| Chemical compound, drug | BrdU | Sigma-Aldrich | B5002 | final concentration of 0,4 g/l |
| Chemical compound, drug | IdU | Sigma-Aldrich | I7125 | final concentration of 0,4 g/l |
| Chemical compound, drug | Trizol | | | |
| Software, algorithm | Ensemble | Public | | |
| Other | DAPI | Roth | | final concentration of 5 ug/l |
| Software, algorithm | DAVID 6.8 | https://david.ncifcrf.gov/home.jsp | | |

## Fish stocks

Wild type and transgenic *Oryzias latipes* (medaka) stocks were maintained in a fish facility built according to the local animal welfare standards (Tierschutzgesetz §11, Abs. 1, Nr. 1). Animal handling and was performed in accordance with European Union animal welfare guidelines and with the approval from the Institutional Animal Care and Use Committees of the National Institute for Basic Biology, Japan. The Heidelberg facility is under the supervision of the local representative of the animal welfare agency. Fish were maintained in a constant recirculating system at 28°C with a 14 hr light/10 hr dark cycle (Tierschutzgesetz 111, Abs. 1, Nr. 1, Haltungserlaubnis AZ35–9185.64 and AZ35–9185.64/BH KIT). The wild-type strain used in this study is Cab, a medaka Southern population strain. We used the the $p53^{E241X}$ mutant (*Taniguchi et al., 2006*) and the following transgenic lines that belong to the Gaudí living toolkit (*Centanin et al., 2014*): Gaudí$^{Ubiq.iCre}$, Gaudí$^{Hsp70.A}$, Gaudí$^{loxP.OUT}$, Gaudí$^{RSG}$ and Gaudí$^{BBW}$.

## Generation of clones and selection of samples for lineage analysis

Clones were generated as previously described (*Centanin et al., 2014*; *Centanin et al., 2011*; *Seleit et al., 2017*; *Rembold et al., 2006*). A brief explanation follows for the different induction protocols. Fish that displayed high recombination were discarded for quantifications on lineage analysis and fate restriction to ensure clonality.

### Inducing recombination via *heat-shock*

Double transgenic Gaudí$^{RSG}$, Gaudí$^{Hsp70.A}$ embryos (stage 20, 24, 29, 32, 34 and 37) were heat-shocked using ERM at 42°C and transferred to 37°C for 1 to 3 hr. Embryos were staged according to *Iwamatsu (2004)*.

### Inducing recombination via tamoxifen

Double transgenic Gaudí$^{RSG}$, Gaudí$^{Ubiq.iCre}$ fish (stage 36 to early juveniles) were placed in a 5 μM Tamoxifen (T5648 Sigma) solution in ERM for 3 hr (short treatment) or 16 hr (long treatment), and

rinsed in abundant fresh ERM before returning them to the plate. Adult fish were placed in a 1 μM Tamoxifen solution in fish water for 4 hr, and washed extensively before returning them to the tank.

## Generating clones via blastula transplantation

Between 25 and 40 cells were transplanted from a labelled, donor blastula to the host, unlabelled blastula. We used a Gaudí$^{loxP.OUT/+}$; p53 $^{\pm}$Gaudí$^{loxP.OUT/+}$ or p53$^{-/-}$ Gaudí$^{loxP.OUT/+}$ as donors, and p53$^{+/-}$, p53$^{-/-}$ and Cabs as hosts. Transplanted embryos were kept in 1xERM supplemented with Penicillin-Streptomycin (Sigma, P0781, used 1/200) and screened for EGFP+ cells in the gills during late embryogenesis.

In all cases, we analysed gills that displayed a recombination efficiency lower than the 25% of stem cells of any pattern, to reduce the chances of independent recombination events in different lineages of the same filament. We annotated whether the 6$^{th}$ filament starting from the peripheral extreme of each branchial arch was labelled or non-labelled, refer this value to the total number of branchial arch extremes (two extremes per branchial arch, and 4 pairs of branchial arches at each side of the gill). We only analysed branchial arches with eight or less stretches of labelled filaments.

## Antibodies and staining protocol

For immunofluorescence stainings we used previously described protocols (*Centanin et al., 2014*). Primary antibodies used in this study were Rabbit a-GFP, Chicken a-GFP (Invitrogen, both 1/750), Rabbit a-Na$^+$K$^+$ATP-ase (Abcam ab76020, EP1845Y, 1/200) and mouse a-BrdU/Idu (Becton Dickinson, 1/50). Secondary antibodies were Alexa 488 a-Rabbit, Alexa 647 a-Rabbit, Alexa 488 a-Chicken (Invitrogen, all 1/500) and Cy5 a-mouse (Jackson, 1/500). DAPI was used in a final concentration of 5 ug/ul.

To stain gills, adult fish were sacrificed using a 2 mg/ml Tricaine solution (Sigma-Aldrich, A5040-25G) and fixed in 4% PFA/PTW for at least 2 hr. Entire Gills were enucleated and fixed overnight in 4% PFA/PTW at 4C, washed extensively with PTW and permeabilised using acetone (10–15 min at −20C). Staining was performed either on entire gills or on separated branchial arches. After staining, samples were transferred to Glycerol 50% and mounted between cover slides using a minimal spacer.

## BrdU or IdU treatment

Stage 41 juveniles were placed in a 0,4 mg/ml BrdU or IdU solution (B5002 and I7125 respectively, Sigma) in ERM for 16 hr and rinsed in abundant fresh ERM before transferring to a tank. Adult fish were placed in a in a 0.4 mg/ml BrdU or IdU solution in fish water for 24 or 48 hr, and washed extensively before returning them to the tank.

## Imaging

Big samples like entire gills or whole branchial arches were imaged under a fluorescent binocular (Olympus MVX10) coupled to a Leica DFC500 camera, or using a Nikon AZ100 scope coupled to a Nikon C1 confocal. Filaments were imaged mostly using confocal Leica TCS SPE, Leica TCS SP8 and Leica TCS SP5 II microscopes. When entire branchial arches were imaged with confocal microscopes, we use the Tile function of a Leica TCS SP8 or a Nikon C2. All image analyses were performed using standard Fiji software.

## RNA-seq data and analysis

Samples were obtained by enucleating gills from five adult medaka fish (between 4 and 6 months old), Cab strain. Apical and medial domains were dissected using scissors, and total RNA was extracted from ca. 30 medial domains and 50 apical domains. Libraries were prepared from total RNA followed by a polyA selection (NEBnext PolyA) and sequenced in a NextSeq 500 platform. They produced an average of 58M 85-nt single end reads for each sample. Each RNA-seq sample was mapped against the oryLat2 assembly using Hisat2 (*Kim et al., 2015*), and the dataset can be accessed at https://www.ncbi.nlm.nih.gov/geo/query/acc.cgi?acc=GSE130939. The aligned read SAM files were assembled into transcripts, their abundance was estimated and tested for differential expression by Cufflinks v2.2.1 (*Trapnell et al., 2012*). Only the expression level variations with both p and q-value less than 0.05 and with the sum of the average FPKM for the same transcript between

the two conditions greater than or equal to 10 were selected. Downstream analysis and graphical representation for Cuffdiff output was done using CummeRbund (*Goff et al., 2018*).

For the GO analysis, we used DAVID (https://david.ncifcrf.gov/) to identify significantly enriched terms associated to the lists of genes differentially expressed at the apical and central domains. To facilitate gene annotations, we obtained the list of zebrafish genes orthologous to that of medaka using Biomart (www.ensembl.org/info/data/biomart/index.html). DAVID analyses were then run using the functional annotation tool with standard parameters (Fisher Exact p-value<0.1).

## Modelling

To model progenitor and stem cell scenarios for the addition of post-embryonic filaments, we performed stochastic simulations for each considering a stretch of 6 filaments, and then compared them to experimental data. We chose stretches of 6 filaments because those guaranteed that we would be focussing on the post-embryonic domain of a branchial arch. A random filament would contain ca. eight embryonic filaments, and we considered branchial arches with 20 or more filaments, which results in six post-embryonic filaments at each side.

### Stem cell model

If there is only one stem cell in the niche, then all six filaments will share the same label, either 0 or 1. We draw random numbers from a Bernoulli distribution, where the probability parameter equals the experimental labelling efficiency of our dataset.

### Progenitor model

In a similar manner, we considered the case of having six progenitor cells in the niche. Thus, this time a Bernoulli process of 6 trials with probability parameter equal to the labeling efficiency of the gill was simulated for each branchial arch.

### Experimental data

We collected data from 22 Gaudí$^{Ubiq.iCRE}$ Gaudí$^{RSG}$ recombined gills, which we dissected and analysed under a confocal microscope and or macroscope - 8 to 16 branchial arches per gill. Subsequently, quantifications were done on the six most peripheral filaments from each side of a branchial arch. The labelling efficiency was estimated for each gill by employing a combinatorial approach: the number of labeled filaments at position +6 (i.e. oldest filaments selected) divided by the total number of branchial arches analysed for that gill.

### Comparison

To compare each model to the experimental data, we compute an objective function in the form of a sum of square differences for each gill and each model. The smaller this objective function is, the better the fit between experimental data and simulations. We annotated both the number of switches and of labelled filaments in each branchial arch.

There exist 19 possible pairs (*s,f*) of switches and labelled filaments, ranging from (0,0), (0,6) up to (5,3). We calculated for each pair *i*, of the form (*s,f*) the frequency of observing it in the data from each gill *j*, $fD_i^{(j)}$, and in simulations of 5000 filament stretches per gill $_jfS_i^{(j)}$. The objective function $f^{(j)}$ was computed for each gill as an adjusted sum of square differences:

$$f^{(j)} = \frac{\sum_{i=1}^{19} \left( fD_i^{(j)} - fS_i^{(j)} \right)^2}{19} \cdot 10^4$$

This was done for both the stem cell and the progenitors models. The factor $10^4$ was introduced for avoiding small numbers thus facilitating the comparison between results. The procedure was repeated 1000 times, producing 1000 objective functions per gill and per model, and therefore obtaining an average value and a standard deviation for each gill for each model.

## Ethics

Experimental procedures with fish were performed in accordance to the German animal welfare law and approved by the local government (Tierschutzgesetz §11, Abs. 1, Nr. 1, husbandry permit number AZ 35–9185.64/BH; line generation permit number AZ 35–9185.81/G-145–15), and with the approval from the Institutional Animal Care and Use Committees of the National Institute for Basic Biology, Japan.

## Acknowledgements

We thank S Lemke, J Wittbrodt, J Lohmann and G Begeman for scientific input at earlier versions of this project, to A Dekanty for feedback on the p53 analysis, and to A Seleit, K Gross and I Krämer for active discussions and suggestions on the manuscript. We are grateful to U Engel and the Nikon Imaging Center for advice and support with microscopes and imaging, to D Ibberson and the Cell Networks Deep Sequencing Core Facility for the preparation of libraries and sequencing, to the NBRP Medaka for fish stocks, and to E Leist, A Sarraceno and M Majewski for fish maintenance. This work has been funded by the Deutsche Forschungsgemeinschaft (German Research Foundation, DFG) via the Collaborative Research Centre SFB873 (subproject A11 to LC and B08 to AMC). LB gratefully acknowledges Ramon Areces Foundation's support, D-PD the Research Training Group (Landesgraduiertenkolleg) "Mathematical Modeling for the Quantitative Biosciences" and Heidelberg Graduate School (HGS MathComp) and LC acknowledges support from the DAAD. JS is the recipient of a Melbourne Research Scholarship from the University of Melbourne, Australia.

## Additional information

### Funding

| Funder | Grant reference number | Author |
|---|---|---|
| University of Melbourne | Melbourne Research Fellowship / Graduate Student Fellowship | Julian Stolper |
| Ramón Areces Foundation | | Lorena Buono |
| University of Heidelberg | | Lorena Buono |
| Deutsche Forschungsgemeinschaft | SFB873/B08 | Anna Marciniak-Czochra |
| Deutsche Forschungsgemeinschaft | SFB873/A11 | Lazaro Centanin |

The funders had no role in study design, data collection and interpretation, or the decision to submit the work for publication.

### Author contributions

Julian Stolper, Elizabeth Mayela Ambrosio, Data curation, Formal analysis, Visualization, Writing—review and editing; Diana-Patricia Danciu, Formal analysis, Methodology, Writing—review and editing; Lorena Buono, Juan R Martínez-Morales, Resources, Data curation, Formal analysis, Visualization, Methodology; David A Elliott, Resources, Writing—review and editing; Kiyoshi Naruse, Resources, Data curation; Anna Marciniak-Czochra, Data curation, Formal analysis, Supervision, Writing—review and editing; Lazaro Centanin, Conceptualization, Data curation, Formal analysis, Supervision, Funding acquisition, Visualization, Writing—original draft

### Author ORCIDs

Elizabeth Mayela Ambrosio (iD) https://orcid.org/0000-0001-7227-7744
Diana-Patricia Danciu (iD) http://orcid.org/0000-0002-8683-3956
David A Elliott (iD) http://orcid.org/0000-0003-1052-7407
Juan R Martínez-Morales (iD) http://orcid.org/0000-0002-4650-4293

Anna Marciniak-Czochra (iD) https://orcid.org/0000-0002-5831-6505
Lazaro Centanin (iD) https://orcid.org/0000-0003-3889-4524

### Ethics

Animal experimentation: Experimental procedures with fish were performed in accordance with the German animal welfare law and approved by the local government (Tierschutzgesetz §11, Abs. 1, Nr. 1, husbandry permit number AZ 35-9185.64/BH; line generation permit number AZ 35-9185.81/G-145-15), and with the approval from the Institutional Animal Care and Use Committees of the National Institute for Basic Biology, Japan.

### Decision letter and Author response

Decision letter https://doi.org/10.7554/eLife.43747.028
Author response https://doi.org/10.7554/eLife.43747.029

## Additional files

### Supplementary files

• Supplementary file 1. Simulation of recombined pattern in branchial arches assuming progenitor cells. The table represents simulated values of recombination (0 = unlabelled; 1 = labelled) in five gills, according to the 'progenitor' model described in M and M. The labelling efficiency was calculated from real data from 5 Gaudí$^{Ubiq.iCre}$ Gaudí$^{RSG}$ recombined gills.
DOI: https://doi.org/10.7554/eLife.43747.018

• Supplementary file 2. Simulation of recombined pattern in branchial arches assuming stem cells. The table represents simulated values of recombination (0 = unlabelled; 1 = labelled) in five gills, according to the 'stem cell' model described in M and M. The labelling efficiency was calculated from real data from 5 Gaudí$^{Ubiq.iCre}$ Gaudí$^{RSG}$ recombined gills.
DOI: https://doi.org/10.7554/eLife.43747.019

• Supplementary file 3. Experimental data of recombined pattern in adult branchial arches. The table includes values for recombination (0 = unlabelled; 1 = labelled) in five gills enucleated from adult Gaudí$^{Ubiq.iCre}$ Gaudí$^{RSG}$ fish that were induced for recombination at late embryonic stages.
DOI: https://doi.org/10.7554/eLife.43747.020

• Supplementary file 4. Objective function representation comparing data to progenitor and stem cell models. The table displays the values for an objective function, comparing the recombination pattern obtained in 22 Gaudí$^{Ubiq.iCre}$ Gaudí$^{RSG}$ recombined gills with the predicted values for a stem cell or a progenitor model (See M and M). Lower values for the objective function represent a better fit between experimental and simulated data.
DOI: https://doi.org/10.7554/eLife.43747.021

• Supplementary file 5. Enriched transcripts in the apical and medial domains of gill filaments. Representation of transcripts obtained from the apical and the medial domain of gill filaments, ordered according to their differential expression.
DOI: https://doi.org/10.7554/eLife.43747.022

• Supplementary file 6. Experimental data of recombined patterns in adult branchial arches with cellular resolution. The table displays values of recombination (0 = unlabelled; 1 = Pattern 1; 2 = Pattern 2; 3 = Pattern 3; 4 = Pattern 4) in gills enucleated from adult Gaudí$^{Ubiq.iCre}$ Gaudí$^{RSG}$ fish that were induced for recombination at late embryonic stages.
DOI: https://doi.org/10.7554/eLife.43747.023

• Transparent reporting form
DOI: https://doi.org/10.7554/eLife.43747.024

### Data availability

All data analysed for this study is included in the manuscript and supporting files. Raw sequencing data have been deposited in GEO under accession code GSE130939.

The following dataset was generated:

| Author(s) | Year | Dataset title | Dataset URL | Database and Identifier |
|---|---|---|---|---|
| Buono L, Martinez-Morales JR, Centanin L | 2019 | Transcriptome analysts of a growth domain and a differentiated domain of the medaka gill | https://www.ncbi.nlm.nih.gov/geo/query/acc.cgi?acc=GSE130939 | NCBI Gene Expression Omnibus, GSE130939 |

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
