## [Decision Letter]

Thank you for submitting your article "Hierarchical stem cell topography splits growth and homeostatic functions in the fish gill" for consideration by *eLife*. Your article has been reviewed by two peer reviewers, and the evaluation has been overseen by Alejandro Sánchez Alvarado as the Reviewing Editor and Didier Stainier as the Senior Editor. The reviewers have opted to remain anonymous.

The reviewers have discussed the reviews with one another and the Reviewing Editor has drafted this decision to help you prepare a revised submission.

Summary:

This carefully executed and richly described effort provides evidence for the existence of a complex stem cell population required for the maintenance and growth of gills in the Japanese rice fish Medaka. The authors have identified a very interesting system in which to study stem cell population dynamics in an adult organ during both injury and homeostasis. By using a combination of lineage tracing and pulse-chase experiments under homeostatic and injury conditions, the authors identified at least 4 unipotent and developmentally restricted adult stem cell types. The documentation of the distinct stem cell systems, in this almost two-dimensional cellular array, is an important finding. In sum, this is a simple and elegant stem cell system that has great potential to uncover novel mechanisms of how one organ system balances growth and regeneration.

Essential revisions:

While all of the reviewers are in agreement of the accessibility and potential of the gill system to inform and uncover dynamics of stem cell regulation during homeostasis and regeneration, there is also agreement that the manuscript in its present form is devoid of molecular detail. As such, the reviewers would appreciate the incorporation of any molecular data the authors may already have that could help address the comments below.

1) The authors describe when the fate of homeostatic and growth aSCs are fixed and locate in the tissue, but they do not show any molecular mechanism of how this is achieved. in situ, qpcr, profiling studies to identify specific stem cells markers in different zone and time points are missing. Could not label retention be used here to identify and isolate these cells and characterize them in some detail?

2) The idea of modifying of the microenvironment is excellent, but it is not clear how specific and accurate the ablation method is. What about the surrounding cells? Controls were needed here.

3) Figures 2 and 4 appear to provide different conclusions. In Figure 2 lineage tracing – the labeled stem cell can give rise to all cells of the filaments. In figure 4, the authors show that in their lineage tracing experiments – 4 distinct unipotent stem cells can give rise to the different patterns observed.

a) Is this difference due to labeling strategy?

b) Why was day 9 chosen for tamoxifen?

c) Does the tip stem cell and other unipotent stem cells arise from a single cell that becomes developmentally restricted?

4) In Figure 4 – the authors suggest that the gill has 4 unipotent stem cells. Evidence supporting that the labeling strategy employed is truly only labeling one cell in their model system is required. This is an important point as these experiments will help determine whether a cell type within the filaments arises from a single labeled cell. Otherwise, how would these specific patterns arise?

5) In Figure 7, the authors ablate filaments and then lineage-trace to determine whether regeneration happens. In the lineage traced animals, the authors identified 4 patterns. Did the authors ablate filaments from all types of lineage-traced patterns? Can all unipotent type of adult stem cells regenerate the tip? Or was the regeneration only seen from pattern 1 animals? Also, is it possible to distinguish between unipotent stem cells driving regeneration or if the differentiated cells re-enter the cell cycle? Are there markers for these cells that can be used for co-staining?

---

## [Author Response]

Essential revisions:While all of the reviewers are in agreement of the accessibility and potential of the gill system to inform and uncover dynamics of stem cell regulation during homeostasis and regeneration, there is also agreement that the manuscript in its present form is devoid of molecular detail. As such, the reviewers would appreciate the incorporation of any molecular data the authors may already have that could help address the comments below.

We are glad to hear that the reviewers appreciate the potential of the gill system, and we understand their concerns about missing molecular data to complement the extensive functional analysis on gill stem cells that we presented in our initial submission. We have improved the revised version by adding two datasets that provide molecular information at different scales.

The first addition complements our functional characterisation of growth stem cell at the tip of the filament. We have conducted RNA-seq on two different regions along gill filaments: the filament trunk, enriched in differentiated cell, and the region that we identified as the filament growth domain, enriched in undifferentiated cells. The analysis of the transcriptomes has allowed us to validate molecularly what we had described functionally. We found numerous genes involved with the physiological function of the gill differentially expressed in the filament trunk, while differentially expressed genes at the growth domain were identified as “*non-differentiated tissue*”. The analysis has been carried out by two colleagues (L.B. and J-R.M.M.), who are included in the author list of the revised manuscript. The data can be found at the new Supplementary file 5, new Figure 3G, new Figure 3—source data 1, and in the new last paragraph of the section “Growth Stem Cells Locate to the Growing Edge of Each Filament”. The raw data has been deposited at: https://www.ncbi.nlm.nih.gov/geo/query/acc.cgi?acc=GSE130939.

The second addition tackles the functional role of the *p53* gene during the coordinated post-embryonic growth of the gill filaments. We have used *p53*/WT chimeras to generate composite filaments, which allowed us to reveal a lineage specific function of P53. Briefly, filaments composed of p53 mutant Pattern 1, 2 and 3 and a WT Pattern 2 display a growth defect when compared to their immediate neighbour filaments with other composition of p53 and WT Patterns. These experiments position p53 as an important regulator of growth stem cells in the fish gill, and reveal a differential contribution of the different lineages to overall growth of the filament. We have included these results in the new Figure 6, new Figure 6—figure supplement 1, and new Section “Wild type – p53-/- chimeras reveal functional differences among lineages”.

We understand that these additional analyses largely complement the main message of our manuscript and want to thank the reviewers for the constructive suggestions.

1) The authors describe when the fate of homeostatic and growth aSCs are fixed and locate in the tissue, but they do not show any molecular mechanism of how this is achieved. in situ, qpcr, profiling studies to identify specific stem cells markers in different zone and time points are missing. Could not label retention be used here to identify and isolate these cells and characterize them in some detail?

We have followed the suggestion of the reviewers to better characterise molecularly gill stem cells, and include in our new version an RNA-seq analysis of different domains along the filaments (new Supplementary file 5, new Figure 3G, and in the new last paragraph of the section “Growth Stem Cells Locate to the Growing Edge of Each Filament”, see previous response).

We have also tried label retention experiments with BrdU (which were indeed used in the initial submission, Figure 3A, E, F), but these were not useful to detect label retention cells in the growth domain. We believe that this is due to the fact that the filaments contain a very small number of stem cells for each lineage, and their high proliferative activity dilutes any label that stem cells can acquire during the BrdU treatment. We are indeed following a bioinformatic approach to estimate the number of stem cells per filament and per brachial arch (with D-PD and AMC), based on the proportion of cells labelled per lineage when a filament displays a labelled pattern. Preliminary analysis suggest that Patterns 2, 3 and 4 are maintained by a single stem cell (since we always get an all-or-none labelling after recombination approached in filament stem cells), and Pattern 2 must contain between 2 and 3.

The low number of growth stem cells per filament also impacts on our RNA-seq analysis on bulk samples. We have detected an expression profile that matches an undifferentiated domain (new Supplementary file 5 and new last paragraph of the section “Growth Stem Cells Locate to the Growing Edge of Each Filament”), but it is clear that a bulk analysis on such sample will most likely reflect the transcriptional profile of amplifying progenitors rather than the one of bona-fide stem cells. Further analysis of single-cell RNA-seq will certainly improve our understanding on molecular pathways controlling growth in each lineage and growth coordination among lineages, and provide a narrower number of candidate genes to test functionally. We believe the work we present here, revealing three different levels of stem cells and 4 types of fate-restricted lineages for each level, constitutes a necessary platform for future, detailed molecular analysis of individual lineages in a constantly growing composite organ. The new, exciting results that we provide using existing p53 mutants (new Figure 6, new Figure 6—figure supplement 1, and new Section “Wild type – p53-/- chimeras reveal functional differences among lineages”) prove the potential of the gill system to understand cross-lineage communication and identify a main factor involved in this coordination.

2) The idea of modifying of the microenvironment is excellent, but it is not clear how specific and accurate the ablation method is. What about the surrounding cells? Controls were needed here.

We understand that the ablation experiments are ill-described in the current version of our manuscript and added the relevant additional information for the revised version, which we believe helps the reader understanding the system better. Filaments are ablated using small surgery scissors on anaesthetised fish, accessing the gill by opening the operculum with dedicated forceps. Filaments are densely packed along a branchial arch, but they conform detached units all along their longitudinal axis. This means that filaments can be treated independently, and the ablation of one of them does not affect the microenvironment on the growth region of neighbour filaments.

We used for this experiment internal controls, i.e. neighbour non-ablated filaments along the same branchial arch, and also the corresponding filaments from the contralateral branchial arch. Both types of control filaments keep growing at the pace of filaments in a control gill, which can be assessed by comparing their relative size along the branchial arch.

To give more visibility to this aspect, we have included new sentences in the Discussion, where we stress the fact that filament growth is an autonomous – i.e., a filament growth rate does not adjust to the size of its neighbour filaments. We believe that this observation is particularly intriguing considering the order in filament size along a branchial arch, where every filament is larger that its younger neighbour and shorter than its older neighbour. Our observations in the regeneration section, and the new results with p53-WT chimeras presented in the revised version, indicate that this stunning distribution is achieved by the autonomous growth of each unit, which positions branchial arches as ideal systems to explore growth coordination – a line of research that we are following and hoping to communicate in the near future.

3) Figures 2 and 4 appear to provide different conclusions. In Figure 2 lineage tracing – the labeled stem cell can give rise to all cells of the filaments. In Figure 4, the authors show that in their lineage tracing experiments – 4 distinct unipotent stem cells can give rise to the different patterns observed.a) Is this difference due to labeling strategy?b) Why was day 9 chosen for tamoxifen?c) Does the tip stem cell and other unipotent stem cells arise from a single cell that becomes developmentally restricted?

We appreciate the comment from the reviewers to help us clarify this important aspect of the post-embryonic growth of the gills. Our results are consistent all along the manuscript and show that filament growth is driven by 4 different fate-restricted stem cells.

The panels in Figure 2 correspond to an entire gill (2B) and en entire branchial arch (2C) that were imaged using a fluorescent binocular, and therefore do not allow characterising clones at the cellular resolution. Figure 2C show filaments labelled in the Pattern1, which is the one comprising the most abundant number of cells, and can be missed for a fully labelled filament. Figure 2B shows filaments labelled in different patterns, but here again the patterns containing more labelled cells (Pattern1 and Pattern2) are the most prominent, easier to detect. The data in Figures 3D, 4, 5, 6, 7, 8 and Figure 5—figure supplement 1 was acquired using confocal microscopes, and they all show lineage restriction. We therefore added a sentence clearly stating this fact in the main text (at the time we describe the results in Figure 2) and in the figure legends to avoid confusions during the initial characterisation of stem cell domains.

In addition, and to answer the points a and c, we include in the revised version additional data using alternative methods of labelling that result in the very same scenario: at least 4 different types of stem cells maintaining growth or each filament and each branchial arch during medaka post-embryonic life. The results presented in the initial version were obtained by inducing recombination on Gaudí*^Ubiq.iCRE^* Gaudí*^RSG^* transgenic fish with tamoxifen at day 9 pf. This developmental stage was chosen as it is the youngest stage of post-embryonic life.

We added now earlier inductions indicating that gills are formed by stem cells that are already fate-restricted before the organ is formed, going against the presence of an initial common stem cell for all lineages in the fish gill. We demonstrate this by using Gaudí*^Hsp70A.CRE^* Gaudí*^RSG^* transgenic fish to induce recombination at different embryonic stages (from stage 20, *ca.* 2 dpf., to stage 34, *ca.* 4 dfp). We have also performed transplantations at the blastula stage, which are arguably the earliest way to generate proper lineage clones in fish, using the truly ubiquitous H2B-EGFP transgenic line Gaudí*^LoxPOUT^*. All these experiments resulted in the same scenario than the previously described, showing the very same recombination patterns in the gill filaments. We have found cases in which all filaments in a branchial arch are labelled in the same pattern (new Figure 5—figure supplement 1B)(Pattern 1 N=1 branchial arch, Pattern 3 N=2 branchial arches, Pattern 4 N=1 branchial arch), indicating that the earliest cell for a given lineage is not contributing to any of the other lineages and these therefore constitute different units. We included this new data in the new Figure 5—figure supplement 1, and in the new last paragraph of the section “Growth Stem Cells are Fate Restricted”.

4) In Figure 4 – the authors suggest that the gill has 4 unipotent stem cells. Evidence supporting that the labeling strategy employed is truly only labeling one cell in their model system is required. This is an important point as these experiments will help determine whether a cell type within the filaments arises from a single labeled cell. Otherwise, how would these specific patterns arise?

This is an important aspect, which we have already partially answered in our previous responses (Answer 1 and Answer 3). In brief, we can not target recombination to one specific filament stem cell in the fish gill. We have, however, tuned tamoxifen concentrations down, reduced the temperature difference during heat-shock inductions, and transplanted fewer cells in our different clone-generating strategies. And in all these samples, we have focused on recombined gills that showed reduced recombination events (this is clearly stated in the main text and the new Materials and methods section of the revised version). We have also included mathematical models to assess the minimal number of cells that are compatible with our experimental observations, and these revealed that the system uses indeed a few number of stem cells per lineage. This is carried out by computing the likelihood of independent recombination events that would result in a stretch of 6 consecutive filaments having the same recombination pattern. We have analysed dozens of gills containing hundreds of labelled filaments and the data obtained adjusts to models that use between 1 and 3 active stem cells per lineage in a given branchial arch (Data presented in Figure 2D and Supplementary files 1-4, 6). In addition, we have used LoxP reporter lines that have additional colours as read outs for recombination (the so-called Brainbow / Confetti principle) and we have observed the same lineage-restriction reported using monochromatic LoxP read-outs. Since these lines contain fluorescent proteins that are not localised to the nucleus, matching these samples to the data we show using H2B-EGFP is non-trivial and that is why we do not include those in the present manuscript.

5) In Figure 7, the authors ablate filaments and then lineage-trace to determine whether regeneration happens. In the lineage traced animals, the authors identified 4 patterns. Did the authors ablate filaments from all types of lineage-traced patterns? Can all unipotent type of adult stem cells regenerate the tip? Or was the regeneration only seen from pattern 1 animals? Also, is it possible to distinguish between unipotent stem cells driving regeneration or if the differentiated cells re-enter the cell cycle? Are there markers for these cells that can be used for co-staining?

We have indeed ablated filaments that contained labelled cells of Pattern 1, Pattern 2 and Pattern 3 (results presented in Figure 8G-I of the revised version). All of them showed the same trend, where recombination patterns are maintained in the new domain formed during the regeneration phase and indicating that the stem cells generating Patters 1-to-3 do not change their fate upon injuries. As previously mentioned, recombination Pattern 4 is difficult to be observed in living samples and this is why we could not perform target ablations on these filaments.

The questions on whether regeneration is triggered by homeostatic stem cells that change to a growth mode, or alternatively, by differentiated cells that re-enter the cell cycle is fascinating and highly relevant in the regeneration field, as both cases have been reported. The ultimate experiment to tackle this question implies using cell-type specific promoters to trigger recombination in post-mitotic cells of the different lineages to then assay whether the regenerative part of a filament derived from labelled or non-labelled cells. We do not have specific markers so far and therefore this particular experiment could not be performed for the revision. We are confident that the use of single-cell RNA-seq will give us valid candidates to approach this aspect for the fish gill. What we can be sure of, is that the regeneration of each individual lineage is dealt with cells of the same lineage. This point is mentioned in the Discussion of the revised version.